



# Sensitivity of the current Antarctic surface mass balance to sea surface conditions using MAR

Christoph Kittel[1], Charles Amory[1], Cécile Agosta[1,2], Alison Delhasse[1], Pierre-Vincent Huot[3], Thierry Fichefet[3], and Xavier Fettweis[1]

[1]Laboratory of Climatology, Department of Geography, University of Liège, Liège, Belgium
[2]Laboratoire des Sciences du Climat et de l'Environnement, Gif-sur-Yvette, France
[3]Earth & Climate, Earth and Life Institute, Catholic University of Louvain, Louvain, Belgium

*Correspondence to:* Christoph Kittel (ckittel@uliege.be)

**Abstract.** Estimates for the recent period and projections of the Antarctic surface mass balance (SMB) often rely on high-resolution polar-oriented regional climate models (RCMs). However, RCMs require large-scale boundary forcing fields provided by reanalyses or general circulation models (GCMs). Since the recent variability of sea surface conditions (SSC, namely sea ice concentration (SIC) and sea surface temperature (SST)) over the Southern Ocean are not reproduced by most GCMs

from the 5th phase of the Coupled Model Intercomparison Project (CMIP5) for the last decades, RCMs are then subject to potential biases. We investigate here the direct sensitivity of the Antarctic SMB to SSC perturbations around the Antarctic. With the RCM MAR, different sensitivity experiments are performed over 1979 – 2015 by altering the ERA-Interim SSC with (i) homogeneous perturbations and (ii) mean anomalies estimated from all CMIP5 models and two extreme ones, while atmospheric lateral boundary conditions remained unchanged. Results show increased (resp. decreased) precipitation due to

perturbations inducing warmer (resp. colder) SSC than ERA-Interim significantly altering the SMB of coastal areas, as precipitation is mainly related to cyclones that do not penetrate far into the continent. At the continental scale, significant SMB anomalies (i.e, greater than the interannual variability) are found for the largest combined SST/SIC perturbations. Sensitivity experiments with warmer SSC reveal integrated SMB anomalies (+5% – +13%) over the present climate (1979 – 2015) in the lower range of the SMB increase projected for the end of the 21st century.

## 1   Introduction

Sea ice concentration (SIC) and sea surface temperature (SST), hereafter referred as sea surface conditions (SSC), influence the exchanges of gases, momentum and heat at the air-sea interface in high latitudes. Due to its high albedo and thermal insulation behaviour, sea ice notably alters the thermodynamic and radiative properties of the ocean surface. Sea ice also prevents evaporation and inherent loading of air masses in water vapour, potentially affecting precipitation at high latitudes.

This is of particular importance for the Antarctic ice sheet (AIS) as the Antarctic surface mass balance (SMB) is mainly controlled by precipitation (van Wessem et al., 2018; Agosta et al., 2018).

Southern Ocean SSC and especially sea ice extent (generally defined as the area of all grid cells with a SIC of at least 15% in satellite and model products) have experienced a large change since the late 1970s (e.g., Massonnet et al., 2013). A



significant increase in sea ice extent has been observed in the Southern Ocean, highly contrasting with the dramatic decline in the Arctic Ocean (Cavalieri and Parkinson, 2012; Parkinson and Cavalieri, 2012). Nonetheless, this general trend conceals major regional differences. For instance, the Amundsen–Bellingshausen Seas showed a strong decrease in sea ice extent, unlike other surrounding Antarctic seas (Turner et al., 2016). Despite the observed changes in the Antarctic SSC and their large

potential impacts on the climate system, the Antarctic SMB did not exhibit any significant trend at the continental scale over the last decades (Bromwich et al., 2011; Lenaerts et al., 2012; Frezzotti et al., 2013; Favier et al., 2017; Agosta et al., 2018).

Several modelling studies have illustrated the influence of open ocean areas on the AIS climate, for instance through a strong atmospheric heating (Simmonds and Budd, 1991; Gallée, 1995), an enhancement of cyclone activity (Simmonds and Wu, 1993; Gallée, 1996; Krinner et al., 2014), and higher precipitation related to intensified evaporation (Wu et al., 1996; Bromwich et al.,

1998; Weatherly, 2004). Conversely, the atmosphere has been shown to be less sensitive to SIC anomalies than SIC to atmosphere anomalies (Simmonds and Jacka, 1995; Bailey and Lynch, 2000) as anomalies induced by the ocean surface are often restricted to the lower atmospheric layers above the Southern Ocean (van Lipzig and van Meijgaard, 2002). However, these previous studies were based on coarse-resolution models (e.g., Weatherly, 2004), with missing physical processes resulting notably in biased surface sublimation (e.g., Noone and Simmonds, 2004), or on regional climate models (RCMs) forced by the

former ERA-15 reanalysis (known as less reliable than more recent renalyses; see (Bromwich et al., 2007)) and run over short periods (van Lipzig and van Meijgaard, 2002).

High-resolution polar-oriented RCMs provide more reliable estimates of the Antarctic SMB components, but they depend on their forcing boundary conditions, including SSC. Using adequate SSC in climate models could be as crucial as using a suitable downscaling model (Krinner et al., 2008; Beaumet et al., 2017). This is of particular importance since most general

circulation models (GCMs) from the 5th phase of the Coupled Model Intercomparison Project (CMIP5; Taylor et al., 2012) fail to reproduce the SSC temporal and spatial variability in the Southern Ocean area over the last decades (Mahlstein et al., 2013; Turner et al., 2013; Shu et al., 2015; Agosta et al., 2015; Roach et al., 2018). As the projected Antarctic SMB could highly depend on the representation of present sea-ice extent (Agosta et al., 2015), we investigate here the sensitivity of the Antarctic SMB to SSC and more specifically to CMIP5 SSC anomalies with the RCM MAR («Modèle Atmosphérique Régional») for

the recent period 1979 – 2015. This will help partitioning the uncertainty in Antarctic SMB projections resulting from biased SSC in GCMs from the uncertainty resulting from biased large-scale circulation patterns. Even tough MAR is a well adapted tool to study the climate sensitivity to SSC (Gallée, 1995, 1996; Messager et al., 2004; Noël et al., 2014), our study only focuses on discussing the direct and local impact of SSC on the Antarctic SMB. This means that we do not consider feedbacks on the general circulation associated to sea ice removal (e.g., Bromwich et al., 1998; Krinner et al., 2014), neither feedbacks

involving sea ice and ocean, but only directs impacts on air temperature, moisture, and SMB components knowing that the general atmospheric circulation remains unchanged in our sensitivity experiments.

A description of MAR, model set-up, and sensitivity experiments is given in Section 2. Section 3 presents model evaluation, as well as the observations and methods used for comparison. The influence of SSC on the Antarctic SMB is analysed in Section 4, while we discuss the direct impacts of altered SSC in Section 5. Our main conclusions are summarized in Section 6.

## 2    Methods and data

### 2.1    The MAR model

The ability of the regional climate model MAR to reproduce the climate specificities of polar regions has been extensively evaluated (e.g., Lang et al., 2015; Gallée et al., 2015; Amory et al., 2015; Fettweis et al., 2017; Agosta et al., 2018). MAR is

a hydrostatic, primitive equation atmospheric model (Gallée and Schayes, 1994) which includes a cloud microphysics module solving conservation equations for specific humidity, cloud droplets, rain drops, cloud ice crystals, and snow particles (Gallée, 1995). Sea spray effects on heat fluxes or on water vapour concentration are parameterized following Andreas (2004). The atmospheric model is coupled to the 1-D SISVAT (Soil Ice Snow Vegetation Atmosphere Transfer; De Ridder and Gallée, 1998) module, which consists of soil and vegetation (De Ridder and Schayes, 1997), snow (Gallée and Duynkerke, 1997;

Gallée et al., 2001) and ice (Lefebre et al., 2003) sub-modules. They simulates energy and mass fluxes between the surface and the atmosphere. The snow-ice module includes sub-modules for surface albedo, melt-water refreezing and snow metamorphism based on the CROCUS model (Brun et al., 1992). Regarding interactions between the atmosphere and ocean, SISVAT considers distinct sea ice and open-water sub-pixels. Open-ocean roughness length for momentum and heat follows Wang (2001), while the momentum roughness length over snow surfaces (sea ice and ice sheet pixels) is computed as a function of air temperature

as proposed in Amory et al. (2017). Consequently, fluxes and roughness lengths are separately calculated over sea ice and water and are afterward weighted according to sea ice and open-ocean proportions (Gallée, 1996). In this study, we use MAR version 3.6.4, recently adapted to Antarctica (Agosta et al., 2018). Although MAR includes a drifting snow module (Gallée et al., 2001), this module has been switched off as in Agosta et al. (2018).

### 2.2    Set-up

As the ERA-Interim reanalysis (Dee et al., 2011) is considered as one of the most reliable reanalyses for the Antarctic region (e.g., Bromwich et al., 2011; Bracegirdle and Marshall, 2012; Agosta et al., 2015), MAR is forced with ERA-Interim every 6 hours over 1979 – 2015 at its atmospheric lateral and upper boundaries (pressure, wind, specific humidity and temperature at each vertical level) and over the ocean surface (SIC and SST). It is worth noting that ERA-Interim uses the SST and SIC values from ERA-40, which are based on monthly and weekly ocean forcing fields (Fiorino, 2004), until January 2002 and

from the OSTIA real daily products afterwards (Stark et al., 2007; Donlon et al., 2012). For each grid cell with a ERA-Interim SIC value greater than 0%, MAR sea ice thickness is initially fixed at 55 cm and sea ice can be covered by snow. The sea ice thickness can then evolve in function of accumulated snowfall or surface snow/ice melt, with a minimum thickness of 10 cm as long as the ERA-Interim SIC is positive. The sub-pixel SST beneath sea ice is fixed at -2 °C in the MAR snow model while the sea ice surface temperature is free to evolve according to its surface energy balance. Finally, as for spin-up time, we starts

our simulations in March 1976 using ERA-40 reanalysis (Uppala et al., 2005) until 1979 with initial snowpack conditions interpolated from previous reference simulations (Agosta et al., 2018).

Compared to Agosta et al. (2018), our integration domain (Fig. 1a) has been extended to include the maximum seasonal sea ice extent as well as major moisture source areas for precipitation particles over the AIS (Sodemann and Stohl, 2009). A

resolution of 50 km has been chosen to preserve a reasonable computational time. The Antarctic topography is based on the digital elevation model Bedmap2 from Fretwell et al. (2013). An upper-air relaxation until 6 km above the surface is used in order to constrain the MAR general atmospheric circulation (van de Berg and Medley, 2016; Agosta et al., 2018). This upper relaxation prevents potential feedbacks between the ocean state and the general atmospheric circulation. Similarly to Noël et al.

(2014), the sensitivity of the Antarctic SMB in our study will thus be restrained to the direct and local impacts of SST and SIC anomalies into the MAR integration domain.

## 2.3   Simulations

In this study, we consider MAR forced by ERA-Interim over 1979 – 2015 as the reference simulation. We perform two sets of sensitivity experiments in which SSC from ERA-Interim are modified as described below. The first set follows the methods

described in Noël et al. (2014), that are simplified and idealized scenarios, while in the second set, SSC are modified according to SSC anomalies from CMIP5 models. In both cases, we analyse the direct impact of SSC anomalies on the Antarctic SMB.

### 2.3.1   SST sensitivity experiments

In these experiments, the 6-hourly ERA-Interim SST are decreased (resp. increased) by 2 °C (SST±2) and 4 °C (SST±4) in ice-free pixels. In cases of an SST reduction, ice-free oceanic pixels are converted into a full ice-covered pixels if the SST drop

below the assumed seawater freezing point (-2 °C) while for an SST increase, the SST of any grid cell with a positive SIC value is limited to the melting point (0 °C) to avoid positive SST and to prevent any SIC change.

### 2.3.2   SIC sensitivity experiments

To prevent too strong changes at the edge between ice-covered and ice-free pixels, ERA-Interim SIC are reduced (resp. increased) by the minimum (resp. maximum) SIC value of the three and six ocean neighbours of each MAR pixel. These ex-

periments are called SIC±3 and SIC±6. Knowing that the resolution is 50 km, this means that the SIC is gradually decreased (resp. increased) by a distance of 150 and 300 km. Following Noël et al. (2014), a SST correction is applied in order to prevent open-water temperature from dropping below -2 °C and SST of sea ice-covered pixels higher than the melting point (0 °C).

### 2.3.3   Combined SST/SIC sensitivity experiments

Combined SST-SIC forcing fields are computed according to the two previous subsections. The added value of these experi-

ments is the simultaneous representation of the increase (resp. decrease) in sea ice extension associated to the decrease (resp. increase) in SST. They are named SST±2/SIC∓3, SST±4/SIC∓6.

### 2.3.4   CMIP5-based sensitivity experiments

In addition to the spatially homogeneous perturbations described above, we evaluate how SSC anomalies from CMIP5 models over the current climate can influence the present-climate Antarctic SMB modelled by RCMs. For that purpose, we interpolate





**Figure 1.** Top: SST anomalies (°C) of (a) NorESM1-ME, (b) CMIP5 average and (c) GISS-E2 compared to ERA-Interim SST over 1979 – 2005. Bottom: SIC anomalies (%) for (d) NorESM1-ME, (e) CMIP5 average and (f) GISS-E2 compared to ERA-Interim SIC over 1979 – 2005. These anomalies were introduced in the 6-hourly ERA-Interim SSC.

mean ocean monthly outputs from all the CMIP5 models using the historical scenario (1979 – 2005) and ERA-Interim by a four-nearest inverse distance-weighted method on our 50 x 50 km MAR grid. The interpolated CMIP5 SIC outputs are compared to the interpolated ERA-Interim SIC fields, considered here as the reference, in order to obtain monthly SIC anomalies. Then, monthly SST anomalies are computed only for grid cells where the SIC value for both interpolated ERA-Interim and CMIP5 model outputs is less than 50%. Finally, SIC and SST anomalies are averaged over 1979 – 2005.

New 6-hourly forcing SST are computed as the sum of ERA-Interim forcing fields and the average of all CMIP5 SST anomalies, hereafter referred to as SST(CMIP5) experiment (Fig. 1c). In the same way, we define SIC(CMIP5) experiments in which SIC anomalies from all CMIP5 models are averaged and applied to the 6-hourly ERA-Interim SIC (Fig. 1g). Introducing CMIP5 anomalies to the original ERA-Interim SSC enables to take into account constant CMIP5 anomalies with the seasonal and inter-annual SSC variability represented in the ERA-Interim reanalysis. The combined SST-SIC anomaly





**Table 1.** JJA and DJF sea ice area (SIA) ($10^6$ km$^2$) within the MAR domain over the period 1979-2015. SIA is defined as the sum of the products of the SIC and area of all grid cells with a SIC value of at least 15%. The DJF (resp. JJA) seasonal mean SST are computed for the ocean free of ice in all experiments in DJF (resp. JJA). We only considered grid cells remaining free of ice (SIC < 15 %) in all experiments in order to remove the influence of sea ice on surface temperature and numerical artefacts due to differences in open ocean areas.

| Experiment | JJA SIA ($10^6$km$^2$) | | DJF SIA ($10^6$km$^2$) | | JJA SST (°C) | | DJF SST (°C) | |
|---|---|---|---|---|---|---|---|---|
| | Mean | Anomaly | Mean | Anomaly | Mean | Anomaly | Mean | Anomaly |
| Reference | 13.31 | | 4.49 | | 5.55 | | 6.36 | |
| SST-4/SIC+6 | 20.63 | +7.32 | 10.83 | +6.34 | 1.55 | -4.00 | 2.36 | -4.00 |
| SST-2/SIC+3 | 17.04 | +3.73 | 7.70 | +3.21 | 3.55 | -2.00 | 4.36 | -2.00 |
| SST+2/SIC-3 | 9.70 | -3.61 | 2.08 | -2.41 | 7.55 | +2.00 | 8.36 | +2.00 |
| SST+4/SIC-6 | 6.77 | -6.54 | 0.96 | -3.53 | 9.55 | +4.00 | 10.36 | +4.00 |
| SST/SIC(NorESM1-ME) | 16.06 | +2.75 | 8.63 | +4.14 | 5.02 | -0.33 | 5.78 | -0.58 |
| SST/SIC(CMIP5) | 12.71 | -0.60 | 4.05 | -0.44 | 5.86 | +0.31 | 6.77 | +0.41 |
| SST/SIC(GISS-E2-H) | 9.66 | -3.65 | 2.34 | -2.15 | 8.30 | +2.75 | 9.22 | +2.86 |

experiment is performed by adding CMIP5 averaged SST and SIC anomalies to ERA-Interim and is hereafter referred to as SST/SIC(CMIP5). Following the same method, we perform combined experiments for two selected CMIP5 models, namely NorESM1-ME (Bentsen et al., 2012) and GISS-E2-H (Schmidt et al., 2014), respectively representative of a colder and warmer ocean than ERA-Interim (Agosta et al., 2015). These experiments are hereafter called SST/SIC(NorESM1-ME) (Fig. 1b,f) and
5 SST/SIC(GISS-E2-H) (Fig. 1d,h).

Table 1 compares SSC perturbations to the reference SSC for June-July-August (JJA) and December-January-February (DJF) SST and sea ice area (SIA). The SIA is defined as the sum of the products of the SIC and area of all grid cells with a SIC value of at least 15%. SIA is preferred to sea ice extent because it better accounts for SIC variations (Roach et al., 2018). Sensitivity experiments with altered SST by ±2 °C and SIC with the ±3 neighbour pixels are in the range of CMIP5 anomalies.
Other perturbations (SST±4 and SIC±6) represent an 1.5 times larger anomaly in SIA and/or SST for both JJA and DJF mean values than CMIP5 mean anomalies over the current climate.

## 3 Evaluation against SMB observations

Since the MAR SMB has already been evaluated against the GLACIOCLIM-SAMBA dataset (Favier et al., 2013) over the AIS at 35 km resolution by Agosta et al. (2018), only a short evaluation is proposed here to highlight a possible deterioration
in the representation of the SMB arising from the use of a coarser resolution. We follow the same method as the one described in Agosta et al. (2018).



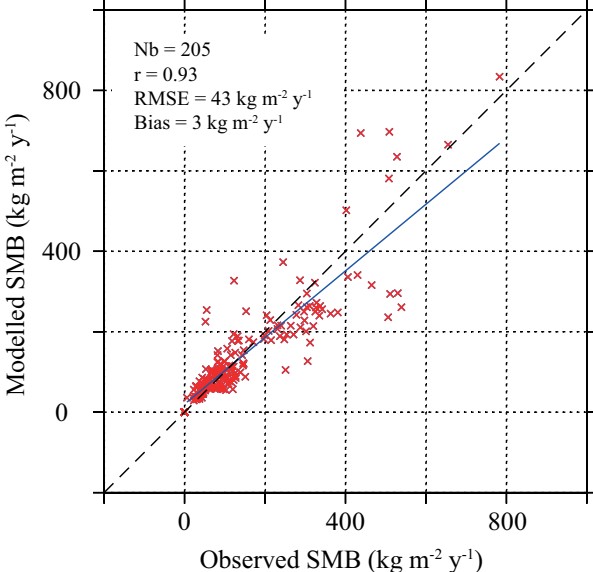

**Figure 2.** Comparison between MAR SMB and observed SMB from the GLACIOCLIM-SAMBA database (Favier et al., 2013) for 1950 – 2015. Bias and RMSE units are kg m$^{-2}$ yr$^{-1}$. The averaged observation mean is 65 kg m$^{-2}$ yr$^{-1}$.

Modelled values are interpolated to observation locations by a four-nearest inverse distance-weighted method. Only SMB observations after 1950 are considered. Concerning observations beginning before our study period (1979 – 2015), the mean 1979 – 2015 modelled SMB values are compared to observations covering more than eight years. For SMB observations beginning after 1979, modelled SMB are compared to SMB observations on the same period. This procedure removed 206
observations from the GLACIOCLIM-SAMBA dataset. Then, all remaining observations located into a same MAR grid cell are averaged, and so are the modelled SMB values previously interpolated to observation locations. Finally, as snow accumulation exhibits a very high variability at the kilometre-scale (Agosta et al., 2012) that cannot be resolved at 50 km resolution, we restrain our comparison to grid cells containing more than one observation, consequently 205 averaged comparison pairs. The comparison without the minimum observation number criterion per grid cell (462 points) is available in supplementary material
(Fig. S1).

The high value of the correlation coefficient (r=0.93) between observed and modelled SMB values shows that MAR correctly represents the Antarctic SMB variability at 50 km resolution over the 1979 – 2015 period, in spite of an overall underestimation of the SMB by 5% on average (Fig. 2). However, it should be noticed that, in the reference run, MAR generally tends to slightly underestimate the SMB values between 200 and 600 kg m$^{-2}$ yr$^{-1}$. These values are typical of accumulation rates at the ice
sheet margins (Agosta et al., 2018). As also shown by Franco et al. (2012) over the Greenland ice sheet, these biases could arise from the coarse resolution used here (50 km) that results in a topography significantly smoothed at the ice sheet margins. This leads to an unsatisfactory representation of the topographic barrier effect allowing the precipitation systems modelled by MAR to penetrate too far inland.



**Table 2.** Top: Annual mean integrated (Gt yr$^{-1}$) and standard deviation (Gt yr$^{-1}$) SMB, precipitation, water fluxes (sublimation and deposition processes) and surface meltwater production over the whole AIS (including grounded and not grounded ice) for the reference run (1979–2015). Positive water fluxes represent a mass loss through sublimation and evaporation while negative water fluxes are representative of deposition processes. Bottom: Difference of annual mean integrated SMB (Gt yr$^{-1}$ and %), its components and meltwater (Gt yr$^{-1}$) between each sensitivity test and the reference simulation (1979 – 2015). Anomalies larger than the inter-annual variability are considered as significant and are displayed in bold.

| Mean (Gt y$^{-1}$) | SMB | | Precipitation | Water fluxes | Meltwater |
|---|---|---|---|---|---|
| Reference | 2569 ± 115 | | 2678 ± 110 | 109 ± 10 | 97 ± 29 |
| Anomaly (Gt y$^{-1}$) | SMB | SMB% | Precipitation | Water fluxes | Meltwater |
| SST-4 | -50 | -1.9 | -64 | **-14** | -21 |
| SST-2 | -82 | -3.2 | -89 | -7 | -21 |
| SST+2 | +41 | +1.6 | +50 | +9 | **+39** |
| SST+4 | **+143** | **+5.6** | **+162** | **+17** | **+117** |
| SIC+6 | **-169** | **-6.6** | **-170** | 0 | -1 |
| SIC+3 | -108 | -4.2 | -107 | +1 | -1 |
| SIC-3 | +24 | +0.9 | +25 | +2 | -5 |
| SIC-6 | +90 | +3.5 | +91 | +1 | -5 |
| SST-4/SIC+6 | **-121** | -4.7 | **-136** | **-15** | +1 |
| SST-2/SIC+3 | **-126** | -4.9 | **-129** | -7 | -11 |
| SST+2/SIC-3 | **+122** | +4.7 | **+133** | +9 | **+53** |
| SST+4/SIC-6 | **+326** | **+12.7** | **+344** | **+13** | **+218** |
| SST/SIC(NorESM1-ME) | -104 | -4.0 | -105 | 0 | +3 |
| SIC(CMIP5) | +36 | +1.4 | +36 | 0 | +7 |
| SST(CMIP5) | +78 | +3.0 | +80 | +1 | +12 |
| SST/SIC(CMIP5) | +103 | +4.0 | +105 | +1 | +18 |
| SST/SIC(GISS-E2-H) | **+355** | **+13.8** | **+368** | **+11** | **+95** |

## 4 Results

In this section, we analyse the local and direct impact of SSC anomalies on the Antarctic SMB and its components modelled by MAR forced by ERA-Interim over 1979 – 2015 (maps of SMB components for all experiments can be found in supplementary material Figs. S2-7). Since liquid precipitation accounts for negligible mass gains compared to snowfall (Table S1), we do no distinctly analyse snowfall and rainfall over the AIS. As the large majority of surface meltwater and rainfall percolates and refreezes into the snowpack, runoff is a negligible component of the Antarctic SMB in both the reference and sensitivity





experiments. However, some runoff events can occur on the Antarctic Peninsula (AP) and are stronger in sensitivity experiments
with warmer SSC (+2 Gt yr$^{-1}$ in SST/SIC(GISS-E2-H) and +6 Gt yr$^{-1}$ in SST+4/SIC-6). The AP is characterized by a sharp
elevation gradient inadequately resolved at 50 km resolution leading to a poor representation of specific climatic processes
encountered in complex topography such as the Foehn effect. Elsewhere coastal runoff amounts stand for very low values. Due

to the coarse model resolution limiting the representation of the atmosphere dynamics over the AP and the marginal contribution
of runoff to surface mass loss compared to sublimation, surface meltwater production is discussed hereafter instead of runoff
amounts. This will help to locate possible areas where the occurrence of surface melting could possibly affect the surface
climate through an increase in snowpack cohesion inhibiting wind erosion (Li and Pomeroy, 1997), or the ice sheet dynamics
through meltwater percolation and subsequent ice shelf destabilization (Bevan et al., 2017).

**4.1   Sensitivity to SST perturbations**

The higher evaporation and air capacity to hold water vapour in SST+2 and SST+4 experiments induce significantly stronger
precipitation rates (i.e., greater than the inter-annual variability) in coastal areas. Figure 3a points out an opposite pattern
between AIS coastal and central areas with a decrease in precipitation over the plateau and large ice shelves (Filchner-Ronne,
Ross and Amery). The warmer ocean lead to an increase in near-surface air temperature of the same magnitude than the

increase in SST converting snowfall in favour to rainfall over the ocean (Fig. S4a,m and Fig. S5a,m). Higher air temperature
also cause a significant increase in surface melt, twice as large as for SST+4 than in the reference simulation. However, melt
and rainfall water can percolate into the snowpack which remains unsaturated except in scarce places. As a consequence, the
surface albedo remains high and does not strengthen melting. Even if mass losses due to surface sublimation are stronger in
SST+4 because of higher air temperature, the SMB anomaly is significantly positive as precipitation changes dominate (Table

20   2).

Conversely, a reduction of the SST leads to non-significant negative integrated SMB anomalies (Table 2). Lower SST weaken
evaporation at the ocean surface and the water vapour holding capacity of the air resulting in smaller annual mean integrated
precipitation over the whole AIS. This decrease in precipitation mainly explains local negative SMB anomalies in coastal areas
(e.g., Victoria Land, Wilkes Land, Drauning Maud Land, Ellsworth Land and Marie Byrd Land, Fig. 3e and Fig. S2p; Fig. S8

locates these coastal areas). However, precipitation over large ice shelves are slightly enhanced and are locally significantly
larger. Over the plateau, stronger deposition processes in addition to a snowfall increase induce a higher SMB than the reference
simulation. These features are discussed in more details in Section 4.3. The total accumulation by deposition is the highest in
SST-4 (Table 2). Moreover, a significant part of rainfall is converted into snowfall over the colder ocean as the near-surface air
is also cooled by the decreased SST (Fig. S4e,p and Fig. S5e,p).

In the SST(CMIP5) experiment (Fig. 3i)), SST are slightly higher (+0.3 °C in winter and +0.4 °C in summer) revealing a
similar pattern than SST+2 and SST+4 even tough non-significant for both integrated and local mean SMB values.





## 4.2 Sensitivity to SIC perturbations

A sea ice retreat induces a precipitation increase over the ice sheet although most of the changes are lower than the interannual variability (Fig. 3b and Fig. S2n) because it favours the advection of moister air masses towards the AIS as already suggested by Gallée (1996). On the opposite, a sea ice increase produces a negative SMB anomaly driven by the reduction of precipitation

over the whole AIS (Fig. 3f, Fig. S2q, and Table 2). Similarly, a significant decrease in the precipitation amounts is observed over the new frozen ocean in SIC+6 (Fig. S3.f) because sea ice mainly acts as an isolator preventing evaporation at the ocean surface. Despite the decrease of the mean summer 2-m air temperature of 10 °C over new sea ice-covered areas in SIC+3 and SIC+6, surface melting does not exhibit a significant decrease over the ice sheet. The sensitivity of the Antarctic SMB to a decrease in SIC seems to be less strong than the sensitivity to an increase in SIC (resp. +3.5% vs. -6.6% for SIC-6/SIC+6),

likely linked to the magnitude of the SIC retreat in SIC-6 and SIC-3 smaller than the magnitude of the sea ice extension in SIC+3 and SIC+6 (Table 1). Finally, the application of CMIP5 SIC pertubations to ERA-Interim fields does not significantly alter both spatial (Fig. 3j) and integrated SMB and its components over the whole AIS (Table 2).

## 4.3 Sensitivity to combined SST/SIC perturbations

Higher SST associated to lower SIC reinforce anomalies found for individual perturbations. Evaporation at the ocean surface

is stronger while warmer air masses have a greater moisture content. Anomalies for integrated precipitation are significantly positive and account for +4.7% and +12.7% in the integrated Antarctic SMB for respectively SST+2/SCI-3 and SST+4/SIC-6 (Table 2). Similarly to SST+4 and SST+2, SST+4/SIC-6 and SST+2/SIC-3 show a large conversion of snowfall to rainfall over the ocean and enhanced precipitation rates over near-coastal regions, while the interior of the AIS exhibits lower accumulation rates (Fig. 3c and Fig. S2o). Moreover, snowfall decreases over the AP but is largely compensated by rainfall refreezing into the

snowpack. Finally, due to higher air temperatures, surface melting and sublimation are also significantly larger. On the opposite, colder SSC (SST-4/SIC+6 and SST-2/SIC+3) prevent evaporation and result in lower precipitation over the AIS (Table 2), more particularly at the ice sheet margins (Fig. 3g and Fig. S2r). While SSC combined sensitivity experiments over the Greenland ice sheet showed similar anomalies than the SST sensitivity experiments (Noël et al., 2014), coupled perturbations act here together to induce stronger anomalies over the AIS than the SST sensitivity experiments. Besides, the AIS sensitivity to SSC

seems to be non-linear for SST, SIC and combined SST-SIC forcing changes, illustrating the complexity of the sea ice/ocean – near-surface atmosphere interactions.

    As SSC anomalies in SST/SIC(GISS-E2-H) are close in magnitude to anomalies in SST+2/SCI-3 and SST+4/SIC-6, integrated values (Table 2) and spatial anomaly patterns (Fig. 3d) also illustrate the positive effect of warmer SSC on the Antarctic SMB by a significant increase in precipitation, sublimation, and melt. However, the spatial pattern of precipita-

tion is slightly different in comparison to SST+4/SIC-6. The precipitation anomaly in SST/SIC(GISS-E2-H) is reduced at Adélie Land and George V Land margins in comparison to SST+4/SIC-6 due to the positive SIC anomalies in GISS-E2-H over the Ross and D'Urville Seas (Fig. 1). SST/SIC(CMIP5) displays a non-significant positive anomaly for both integrated and spatial SMB (Fig. 3k) as SIC and SST anomalies in CMIP5 models are mainly equally distributed around the mean. Finally,





for SST/SIC(NorESM1-ME), SSC are representative of a colder ocean (lower SST and essentially higher SIC) resulting in a non-significant negative SMB anomaly with a precipitation decrease at the ice edge and over the external areas of the plateau (Fig. 3h).

Since snowfall is the largest component of the Antarctic SMB, precipitation changes mainly explain the spatial anomaly patterns observed in our experiments. A warmer ocean with a smaller sea ice cover tends to strongly enhance precipitation at the ice sheet margins while decreased accumulation rates are modelled over the ice shelves and the central part of the ice sheet. Since the temperature of the air masses from the central AIS are not impacted by SSC, a higher humidity content at the ice margins results in a quicker saturation of air masses during lift-up processes and the meeting with downward cold air masses from the central AIS as described by Gallée (1996). More precipitation falls then over the steep margins further reducing precipitation inland and over areas downwind of the AP (Fig. S3c). However, it should be noted that the coarse 50 km resolution used here does not adequately resolve the orography leading to artificially overestimated precipitation on the windward sides of orographic barriers. This is notably the case over the Filchner-Ronne and Ross ice shelves where the Amundsen Sea Low generates a return flow. On the contrary, sensitivity experiments based on a coupled increase in SIC and a SST cooling exhibit the opposite pattern. Knowing that air masses are drier, they have to rise up higher to reach saturation so that precipitation systems can form further inland.

## 5 Discussion

Even if our sensitivity experiments rely on larger SSC perturbations than the interannual variability, mean integrated SMB anomalies are not systematically significant in comparison to our reference simulation. As already shown for the Greenland ice sheet (Noël et al., 2014), katabatic winds prevent significant impacts of SSC on the Antarctic SMB. Anomalies in moisture and temperature caused by SSC changes are confined under an inversion layer (Deser et al., 2010) and blocked by katabatic winds, limiting the transport of moisture and energy towards the central part of the AIS. However, SSC perturbations in the range of CMIP5 anomalies modify the Antarctic SMB with integrated anomalies close to the interannual variability, and even three times larger for some models (e.g., GISS-E2-H).

In the context of Global Warming, it is important to note that the Antarctic SMB increases by 2 – 6% for a SST increase alone and by 5 – 13% for a SST increase coupled to a SIC drop (Table 2). Similar increases are found in sensitivity experiments based on CMIP5 SSC anomalies compared to ERA-Interim over the current climate (+4% and +13% respectively for SST/SIC(CMIP5) and SST/SIC(GISS-E2-H)). Knowing that the regional model RACMO2 projects an increase in SMB by 6–16% in 2100 (Ligtenberg et al., 2013) and the global model LMDZ4 suggests a SMB increase of 17% for the same horizon (Krinner et al., 2008), present-climate simulations with altered SSC in the range of present-climate biases from CMIP5 models lead to SMB anomalies over the current climate in the (lower) range of the projected SMB increase for the end of the 21st century.

Sensitivity experiments are compared to SMB observations from the GLACIOCLIM-SAMBA dataset with the same methods used for the reference simulation (Sect. 3). Correlation and RMSE do not vary significantly between the sensitivity experiments





**Figure 3.** Difference in mean annual SMB (kg m$^{-2}$ y$^{-1}$) between the reference simulation and (a) SST+4, (b) SIC-6, (c) SST+4/SIC-6, (d) SST/SIC(GISS-E2-H), (e) SST-4, (f) SIC+6, (g) SST-4/SIC+6, (h) SST/SIC(NorESM1-ME), (i) SST(CMIP5), (i) SIC(CMIP5), (k) SST/SIC(CMIP5) experiments. Difference less than the interannual variability are considered as non-significant and are dashed. l) Mean annual SMB (kg m$^{-2}$ y$^{-1}$) simulated by MAR forced by ERA-Interim over 1979 – 2015.





**Table 3.** Comparison between modelled and observed SMB from the GLACIOCLIM-SAMBA database (Favier et al., 2013) over 1950–2015. Bias et RMSE units are kg m$^{-2}$ yr$^{-1}$. The observation mean is 65 kg m$^{-2}$ yr$^{-1}$ while the observation standard deviation is 119 kg m$^{-2}$ yr$^{-1}$.

| Simulation acronym | SMB (kg m$^{-2}$ yr$^{-1}$) | | |
|---|---|---|---|
| | R | BIAS | RMSE |
| Reference | 0.93 | -3 | 43 |
| SST-4 | 0.93 | -5 | 43 |
| SST-2 | 0.93 | -5 | 43 |
| SST+2 | 0.93 | -2 | 43 |
| SST+4 | 0.93 | +1 | 48 |
| SIC+6 | 0.93 | -8 | 42 |
| SIC+3 | 0.94 | -6 | 43 |
| SIC-3 | 0.93 | -3 | 43 |
| SIC-6 | 0.93 | 0 | 43 |
| SST-4/SIC+6 | 0.93 | -6 | 44 |
| SST-2/SIC+3 | 0.93 | -7 | 44 |
| SST+2/SIC-3 | 0.93 | 0 | 45 |
| SST+4/SIC-6 | 0.92 | +6 | 55 |
| SST/SIC-NorESM1-ME | 0.93 | -7 | 44 |
| SIC-CMIP5 | 0.93 | -3 | 44 |
| SST-CMIP5 | 0.93 | +3 | 47 |
| SST/SIC-CMIP5 | 0.93 | 0 | 43 |
| SST/SIC-GISS-E2-H | 0.93 | +8 | 52 |

and the reference run (Table 3). Only mean biases vary but the variations are far smaller than the observed variability. Consequently, sensitivity experiments with potential large local or integrated SMB anomalies seems to not significantly affect the comparison with observations. This is explained by the low amount of available observations and highlights the importance of continuing to carry out field campaign measurements, as well as extending their spatial coverage to better assess model results.

## 5  6  Conclusion

Polar-oriented RCMs are suitable numerical tools to study the SMB of the AIS due to their high spatial resolution and adapted physics. Nonetheless, they are driven at their atmospheric and oceanic boundaries by reanalyses or GCM products and are then influenced by potential biases in these ones. These biases can be notably significant for SSC (e.g., Agosta et al., 2015). With the





RCM MAR, two sets of sensitivity experiments were carried out to assess the direct response of the Antarctic SMB to oceanic perturbations around Antarctica by altering the ERA-Interim SSC over 1979 – 2015 while keeping unchanged the atmospheric conditions at the MAR lateral and upper boundaries. The first set consisted in spatially homogeneous SSC perturbations. The second set of experiments involved ERA-Interim SSC perturbations estimated from CMIP5 models anomalies over the present climate. We introduced mean anomalies from the historical run of CMIP5 models and two extreme models of CMIP5, namely NorESM1-ME and GISS-E2-H respectively representative of warmer (i.e., higher SST and lower SIC) and colder (i.e., lower SST and higher SIC) SSC than ERA-Interim.

Results mainly show increased (decreased) precipitation due to anomalies related to warmer (colder) SSC affecting the SMB of the AIS. As precipitation is mainly caused by low-pressure systems that intrude into the continent and do not penetrate far inland, coastal areas are more sensitive to SSC perturbations with more significant anomalies compared to inland regions. Warmer SSC significantly enhance precipitation at the ice sheet margins since a greater moisture loading of air masses leads to saturation more rapidly as they rise and adiabatically cool over the topography. On the contrary, colder SSC reduce precipitation at the ice sheet margins and slightly increase it further inland as air masses have to rise up to higher elevations to reach saturation. Finally, the largest combined SST/SIC perturbations lead to significant (i.e, greater than the interannual variability) SMB anomalies integrated over the whole AIS. However, comparing the results of the various modelling experiments with SMB observations shows that statistics remained mostly identical, suggesting that large integrated anomalies can remain unperceived if compared to scarce field observations.

Our sensitivity tests with warmer (CMIP5-based) altered oceans reveal SMB anomalies over the current climate in the lower range of the SMB increase projected for the end of the 21st century. Given the influence of SSC perturbations on the Antarctic SMB over the current climate as demonstrated in this study, a special attention should be paid to future SMB projections using potentially biased SSC. This highlights the necessity of improving the representation of the present-climate SSC in the context of downscaling the forthcoming CMIP6 model outputs to produce new Antarctic SMB projections.

*Competing interests.* The authors declare that they have no conflict of interest.

*Acknowledgements.* Computational resources have been provided by the Consortium des Équipements de Calcul Intensif (CÉCI), funded by the Fonds de la Recherche Scientifique de Belgique (F.R.S. – FNRS) under grant no. 2.5020.11 and the Tier-1 supercomputer (Zenobe) of the Fédération Wallonie Bruxelles infrastructure funded by the Walloon Region under the grant agreement no. 1117545. This work was supported by the Fonds de la Recherche Scientifique - FNRS under Grand no. T.0002.16.



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
