# Peer review of "Sensitivity of the current Antarctic surface mass balance to sea surface conditions using MAR"

_The Cryosphere, 2018_

## Referee Comment (RC1) · Anonymous Referee #1 · 24 Jul 2018

**Review of: "Sensitivity of the current Antarctic surface mass balance to sea surface conditions using MAR", by *C. Kittel et al.*, submitted to *The Cryosphere*.**

Using the Modèle Atmosphérique Régional (MAR), the authors investigate the impact of perturbations in present-day sea surface conditions (SSC), i.e. changes in sea surface temperature (SST) and sea ice cover (SIC), on the surface mass balance (SMB) of the Antarctic Ice Sheet (AIS). To that end, the authors carry out a reference run at 50 km horizontal resolution, forced by ERA-Interim (1979-2015) reanalysis, as well as a set of sensitivity experiments perturbing SSC as prescribed by the reanalysis: SST ± 2-4ºC, SIC ± 3-6 MAR grid-cells and combined SST/SIC perturbations. In addition, the authors also force MAR by SSC prescribed from an ensemble average of CMIP5 models and two extreme members, characterized by colder (GISS-E2-H) and warmer (NorESM1-ME) oceanic conditions relative to the reference run. For these simulations, no feedbacks on the atmospheric conditions are considered to exclusively investigate the direct impact of SSC perturbations on the AIS SMB.

Comparing the reference simulation to these sensitivity experiments, the authors show that integrated precipitation, i.e. major contributor to the AIS SMB, significantly increases (resp. decreases) at the ice sheet margins for "warmer" (resp. "colder") SSC experiments, i.e. higher (resp. lower) SST and reduced (resp. increased) SIC, leading to significantly positive (resp. negative) SMB anomalies. This is even more pronounced for combined SSC perturbations (SST/SIC). This study suggests that AIS SMB is more sensitive to SST perturbations than SIC ones. For CMIP5-forced experiments, small and insignificant SMB anomalies are found compared to the reference run. Finally, the authors stress that SMB anomalies obtained from the most extreme SSC perturbation, i.e. SST+4ºC / SIC-6, stand in the lower range of the projected AIS SMB increase by 2100.

This is a sound study based on a well evaluated, state-of-the-art climate model. The sensitivity experiments are well designed and the use of CMIP5 ensemble and members gives an additional insight on how well GCMs currently resolve SSC, as well as biases to be expected when these modeled SSC are used for future projections of the AIS SMB. The paper is generally well written, and includes clear tables and figures. I deem that **minor revisions** are required before acceptance for publication in the Cryosphere. Hereunder the authors can find my comments and concerns that should be addressed before publication. Stylistic suggestions are also listed below.

**General comments**

1. At several instances, the authors acknowledge that a horizontal resolution of 50 km is inadequate to accurately resolve orographic-forced and local precipitation at the AIS rough margins and over the Antarctic Peninsula (AP): e.g. P7 L17-18, P9 L2-4, P11 L10-13. Although the authors are fair on this point, and justify the use of a coarse spatial resolution as a trade off between manageable computational time and the number of simulation carried out, while still resolving the AIS SMB reasonably well (see Fig. 2), they fail at estimating the associated biases and uncertainties. This is an important concern as most SMB anomalies are found in marginal (steep) regions where the authors suggest potentially large resolution-driven precipitation biases.

   To address this issue, the authors should present a 2D comparison between the 50 km reference run (this study) and the state-of-the-art 35 km run (Agosta et al., 2018), both forced by ERA-Interim (1979-2015). This would highlight the spatial distribution of precipitation/SMB biases and point out where large uncertainties, in both the reference run and sensitivity experiments, are likely to be found. This additional analysis would help the reader interpreting the significance of the SMB anomalies obtained in the sensitivity experiments; in other words, whether these SMB anomalies are larger/smaller than the local difference between the two MAR runs at 50 km and 35 km resolution.

2. As for comparison, the authors should also consider including a second scatterplot in Fig. 2 for the 35 km run, and list the associated statistics. Integrated values and uncertainty (standard deviation) derived from the 35 km run should also be listed in Table 1.

3. An additional Section 6 "Limitations" could discuss in more detail differences in SMB between the 50 km and 35 km simulations as well as related model limitations, i.e. unresolved or not well resolved foehn effect and orographic-enhancement affecting precipitation e.g. over the AP.

4. I would strongly advise to replace Fig. 3 by Fig. S2 in the main manuscript, as it displays all sensitivity experiments performed. Fig. S2 is otherwise redundant, and it is rather inefficient to move back and forth to the supplementary material to visualize and compare results from experiments not shown in the main manuscript.

**Substantive Comments**

**Section 2.3.4: P5 L1-2:** Here, I understand that monthly SSC (1979-2005) derived from CMIP5 ensemble average and two extreme members, as well as from ERA-Interim are interpolated to the MAR grid (50x50 km) using an inverse distance weighting method based on the four CMIP5 models/ERA-Interim cells nearest to the current MAR one. If so, please reformulate accordingly. **L5:** This is confusing, are the authors calculating monthly mean SIC-SST from CMIP5/ERA-Interim for 1979-2005 (12 values) or an annual mean (1 value). I understand that monthly SIC and SST anomalies are used, please clarify. **L6-10:** Are the new 6-hourly SST-SIC calculated as the sum of 6-hourly ERA-Interim (i.e. for a specific day of a certain month) and the corresponding monthly anomaly in SST-SIC from CMIP5 models? If so, please reformulate.

**P7 L17-18:** This is an important caveat that should be addressed in an additional section "Limitations". A 2D map comparison between 50 km and 35 km simulations could help the reader understanding where large uncertainties are likely to be found at 50 km, see also general comments #1 and #3.

**Point Comments**

**P1 L10:** The authors refer to "warm SSC" or "cold SSC" several times across the manuscript. While it may sound obvious that warm (resp. cold) SSC represents combined low SIC and high SST (resp. high SIC and low SST), this should be explicitly stated in the manuscript, e.g. in Section 2.3.3 or more generally in Section 2.3.
**P2 L29-30:** What do the authors mean by "neither feedbacks involving sea ice and ocean"?
**P3 L18:** Could the authors briefly elaborate on the reasons why the drifting snow module is switched off? **L25:** Could the authors define OSTIA? **L31:** I guess the authors mean "[…] from a previous reference simulation", or is the initialization based on multiple simulations, please clarify.
**P4 L2:** Could the authors mention the original resolution of their DEM?
**P5 L7:** I guess the authors mean (Fig. 1b). **L8:** (Fig. 1e).
**P6 L3:** Could the authors estimate by how many ºC on average these two extreme CMIP5 members are "colder" or "warmer" than ERA-Interim. **L4-5:** I guess the authors mean: (Fig. 1a,d) and (Fig. 1c,f.
**P7 L4:** Could the authors mention how many measurements were discarded from the evaluation?
**P9 L2-7:** Modeling limitations in the AP could be discussed in an additional "Limitations" section, see general comment #3.
**P10 L11-12:** Do the authors mean that the SMB from SIC(CMIP5) does not significantly differ from the reference, both spatially and integrated over the whole AIS? If so, please reformulate. **L16:** "SST+2/SIC-3", same at **L27**. **L19-20:** I do not see snowfall decreasing over the AP in the supplementary figures, this rather seems to occur in the surrounding ocean. Please clarify. **L32:** The authors certainly mean (Fig. 1f). **L32-33:** "SST/SIC(CMIP5) suggests", what do the authors mean by "as SIC and SST anomalies […] around the mean"? Please, clarify.
**P11 L7-9:** I am not sure to understand the links between unchanged inland temperature, coastal precipitation enhancement and downward (katabatic?) winds. Could the authors reformulate this

sentence? **L10-13:** This discussion should be moved to a new section "Limitations". **L29-31:** Please reformulate this sentence "present-climate […] 21$^{st}$ century", it reads better at **L18-19 of P14**.

**Stylistic suggestions**

**P1 L2:** […] boundary forcing fields prescribed by reanalyses […]. **L7:** Altering is rather negative, and suggests that data have been deteriorated. I would suggest: "by modifying the ERA-Interim SSC". **L10:** Replace "altering" by "affecting". **L17:** "exchange of gas". **L18:** Remove "behaviour"; "impacts" or "affects" instead of "alters". **L19:** "[…] water vapour loading of air masses, potentially […]". **L20:** Remove "as the Antarctic" and insert "that" after "(SMB)". **L23:** The authors could consider reformulating as follows: "[…] have experienced a significant increase since the 1970s (e.g. Massonnet et al., 2013), highly contrasting with the dramatic decline reported in the Arctic Ocean […]".

**P2 L9:** Maybe "[…] and increased precipitation […]". **L13:** I suggest: "[…] (e.g. Weatherly, 2014), with simplified physics resulting […]". **L14:** I suggest: "[…] (RCMs) forced by former and less reliable reanalyses (e.g. ERA-15 in Bromwich et al., 2007) over short periods […]". **L24:** "[…] with the 'Modèle Atmosphérique Régional' (MAR) for the period 1979-2015. This allows partitioning […]". **L27-31:** I suggest: "[…], this study only discusses the direct […]. This means that no feedback on the […] removal is considered […] Krinner et al. 2014). Only direct impacts on […] components are accounted for. Note that the general […]". **L34:** Replace "altered" by "perturbed".

**P3 L7:** Maybe "The effect of sea spray on […]". **L10:** I suggest: "[…] sub-modules, that simulate energy and […]". **L15:** Remove "Consequently,". **L16:** I suggest: "[…] and open-ocean fractions". **L21:** "forced by". **L27:** "[…] evolve as a function of accumulated snowfall and surface […]". **L29:** "we start". **L33:** "[…] moisture source for precipitation over the AIS […]".

**P4 L1:** Maybe "selected" instead of "chosen". **L2:** "extending 6 km above". **L5:** "[…] SMB to SSC perturbations is limited to the […] SIC anomalies within the MAR […]". **L9-10:** "perturbed" instead of "modified". **L13-16:** "for ice-free pixels", "converted into full ice-covered pixels if the SST drops […] (-2ºC). For an SST […] SIC value is set to […]". **L18:** Maybe "at the interface between […]". **L22:** "[…] and maintain SST of sea ice-covered pixels […]". **L25:** "in sea ice extent associated". **L29:** Replace "present climate" by "present-day" or "contemporary".

**P5 L9:** "[…] anomalies into the original […] enables to account for constant […]".

**P6 L9:** "perturbed" instead of "altered". **L10:** I suggest: "SST and/or SIA anomalies for JJA and DJF periods are 1.5 times as large as CMIP5 mean […]". **L14:** "brief" instead of "short". **L14:** This is rather negative, I would suggest: "to highlight the impact of using a coarser horizontal resolution on SMB representation". **L15:** Remove "the one described".

**P7 L1:** Replace "by" by "using". **L2:** "observations collected prior to our study period". **L3:** "for the same period.". **L6:** Add "the" before "observation locations". **L7:** I suggest: "(Agosta et al., 2012), unresolved at 50 km […]". **L8:** Replace "consequently" by "i.e." or by "or". **L13:** "noted" instead of "noticed". **L16:** "significantly smoothed topography".

**P9 L1:** "enhanced" instead of "stronger". **L7:** "This allows locating". **L12:** For consistency, "interannual variability". **L14:** "The warmer ocean leads to […] of similar magnitude as […] converting snowfall into rainfall." **L16:** "[…] higher temperature also causes […] SST+4 relative to the reference simulation (Fig. S7a-l)." **L18-20:** "[…] sublimation are larger in SST+4 (Fig. S6a) because […] temperature, increased precipitation dominates and the SMB anomaly is significantly positive (Fig. 3a and Table 2)." **L22:** "ocean surface and reduce the water vapour". **L26:** "Over the plateau, larger deposition combined with snowfall […] than for the reference run.". **L28-29:** "[…] colder ocean as lower SST also decrease the near-surface air temperature […]". **L31:** "revealing similar patterns as in SST+2 (Fig. S2m) although non-significant […]".

**P10 L1:** "smaller" instead of "lower". **L5-7:** "[…] decrease in precipitation is observed over the new ice-covered ocean […] act as an insulator […] 2-m air temperature by 10ºC". **L9:** Maybe "pronounced" instead of "strong"; "[…] SIC-6/SIC+6). This is likely due to the smaller magnitude […] SIC-3 compared to the magnitude of SIC extension […]". **L23:** "show similar anomalies as". **L24:** "larger anomalies […] Besides, the sensitivity of AIS SMB to SSC is non-linear".

**P11 L2:** Maybe "marginal" instead of "external". **L14-15:** I suggest: "opposite pattern: drier air masses have to rise up higher […] so that precipitation is generated further inland.". **L22:** "affect" instead of "modify". **L23:** "three times as large". **L29:** "perturbed SSC".

**P13 L1:** "are by far smaller". **L2:** Maybe "showing" instead of "with potential"; the authors could consider "[…] SMB anomalies do not significantly differ from observed SMB". **L4:** Replace "assess" by "evaluate". **L7:** Replace "driven" by "forced".

**P14 L2:** Replace "altering" by "perturbing" and move "unchanged" to the end of the sentence. **L3:** "The first set consists of […]". **L8:** "(resp. decreased) precipitation due to warmer (resp. colder) […]". **L12:** Maybe "leads to earlier/faster saturation as they rise […]". **L15:** I suggest: "However, comparing modelled SMB from sensitivity experiments with observations shows no significant difference, suggesting […]". **L18:** "with warmer (CMIP5-based) SSC reveal that SMB […] climate stand in the lower". **L20:** Remove "as demonstrated in this study". **L21:** "using potentially biased SSC as forcing". **L22:** Replace "produce new" by "carry out future".

**Figures and Tables**

**Table 1:** For the reference run, insert "0.00" or "-" to fill the blanks in the anomaly columns.

**Table 2:** Caption **L2**: "floating ice" instead of "not grounded ice". For comparison with the coarser resolution simulation, the authors should consider adding integrated numbers from the 35 km run discussed in Agosta et al. (2018). See also general comment #2.

**Fig. 2:** For comparison, the authors should consider including a second scatterplot showing outputs of the 35 km run.

**Fig. 3:** This figure could be replaced by Fig. S2. See also general comment #4.

**Fig. S6:** Caption **L1**: "minus" instead of "mines".

---

## Referee Comment (RC2) · Anonymous Referee #2 · 11 Sep 2018

This paper is an interesting analysis of the sensitivity of the Antarctic surface mass balance in a regional climate model to perturbed sea surface conditions (sea surface temperature and sea ice extent). The paper is well written and the results are relevant. There are some mechanisms given in the paper to explain some of the sensitivities, but there are no quantative analyses done to support these hypotheses. I therefore recommend such analyses to be added to the manuscript (see major point 3 and 3) before it can be accepted. 1. You perturb both the SST and the SIC, but not necessarily in a consistent way. In my opinion, more material should be given to illustrate a consist perturbation, for example by comparing how the SST bias in the GCM compares to the SIC bias? Another way would be to assume – for example – that meridional SST gradient remains unchanged as the SST increases, which imposes the retreat of the

sea-ice edge. 2. I do not really following the reasoning throughout the paper why there is more precipitation inland when SST is lower or SIC is higher. You argue that this is because the dryer air has to rise up higher to reach saturation. Although, this is of course true, it does not imply that precipitation can be brought higher up – it just means the saturation point is at a higher elevation. For saturated air, the amount of moisture transported in the interior is only dependent on temperature and circulation. So additional analyses are needed to shed light on this issue. The best way would be to do a moisture budget over the interior and see whether small circulation changes might be responsible for this. Although you do spectral nudging, circulation close to the surface might deviate which can be relevant for moisture advection. Although this comment is valid for the entire results/discussion section, p11, line 14/15 is particularly misleading. 3. On p 11 linr 19 you state that 'Katabatic winds prevent significant impacts of SSC on the Antarctic SMB'. Although this might be true, I do not see proof for this in the manuscript. Even if there would be no katabatics, the fact that air has to rise over the topographic barrier and additional moisture is constrained to the boundary layer, might be enough to prevent significant effect. The paper would need to be strengthened with some quantitative postprocessing of the model output to shed light on this explanation. Minor comments: 1. Abstract: last sentence: a number for a sensitivity in % is meaningless when the magnitude of the perturbation is not specified. Please clarify in the abstract 2. P1, line 22: I am not sure if I follow the definition of the Sea Ice Extent given there. Can you give a reference for this definition or clarify? 3. P 2, l11: reference is van Lipzig et al., (2002) not van Lipzig and van Meijgaard (see below). 4. P9: l12: Air does not have 'a capacity' to hold water vapour. The water vapour is one of the components of air. Please reformulate. Van Lipzig, N.P.M., E. van Meijgaard and J. Oerlemans, 2002. Temperature sensitivity of the Antarctic surface mass balance in a regional atmospheric climate model. J. Clim., 15(19), 2758-2774. doi:10.1175/1520-0442(2002)015<2758:TSOTAS>2.0.CO;2.

---

## Author Response (AR1)

Dear Editor,

We would like to thank both reviewers for their relevant comments which have helped us to improve our manuscript.

Responses to each individual reviewer have been posted on TCD and can be also found hereafter.

The main changes compared to the original version are:

-the addition of Sébastien Doutreloup (ULiège) and Coraline Wyard (Uliège) as co-author to thank them for the constructive discussions and their help with the new figures presented in the revised version of our manuscript.
-the addition of a discussion about the biases of MAR at a 50 km resolution compared to MAR results at 35 km as well as new figures in supplementary material as requested by the reviewer #1 in order to assess the uncertainty related to the coarser resolution.
-a more detailed analysis of processes responsible of the increase in precipitation inland in the sensitivity experiments with a colder ocean as requested by the reviewer#2.

Best regards,
Christoph K. on behalf of the co-authors

We first would like to thank the reviewer #1 for his constructive comments which will help to improve our manuscript.

**General comments**

1. At several instances, the authors acknowledge that a horizontal resolution of 50 km is inadequate to accurately resolve orographic--forced and local precipitation at the AIS rough margins and over the Antarctic Peninsula (AP): e.g. P7 L17--18, P9 L2--4, P11 L10--13. Although the authors are fair on this point, and justify the use of a coarse spatial resolution as a tradeoff between manageable computational time and the number of simulation carried out, while still resolving the AIS SMB reasonably well (see Fig. 2), they fail at estimating the associated biases and uncertainties. This is an important concern as most SMB anomalies are found in marginal (steep) regions where the authors suggest potentially large resolution-- driven precipitation biases.

   To address this issue, the authors should present a 2D comparison between the 50 km reference run (this study) and the state--of--the--art 35 km run (Agosta et al., 2018), both forced by ERA--Interim (1979--2015). This would highlight the spatial distribution of precipitation/SMB biases and point out where large uncertainties, in both the reference run and sensitivity experiments, are likely to be found. This additional analysis would help the reader interpreting the significance of the SMB anomalies obtained in the sensitivity experiments; in other words, whether these SMB anomalies are larger/smaller than the local difference between the two MAR runs at 50 km and 35 km resolution.

2. As for comparison, the authors should also consider including a second scatterplot in Fig. 2 for the 35 km run, and list the associated statistics. Integrated values and uncertainty (standard deviation) derived from the 35 km run should also be listed in Table 1.

3. An additional Section 6 "Limitations" could discuss in more detail differences in SMB between the 50 km and 35 km simulations as well as related model limitations, i.e. unresolved or not well resolved foehn effect and orographic--enhancement affecting precipitation e.g. over the AP.

(Response to general comments 1 to 3)
As highlighted by the reviewer, the influence of the resolution is not discussed in our paper even if we use a coarser spatial resolution than the previous study using MAR (Agosta *et al.*, 2018). However, we think that discussing the sensitivity of the Antarctic simulated SMB to the (spatial) resolution used in the model is beyond the scope of this study. Furthermore, our methods for comparing the modeled and observed SMB will not enable a fair comparison between the statistics for 50km and 35km simulations as the number of pixels used for the comparison differs and becomes very small for the 35km resolution grid if our criterion of observations (P7L8) by pixel is kept (i.e, more than one observation by pixel).

Since the sensitivity of the Antarctic SMB to the horizontal resolution is an interesting matter of debate and still an unanswered question, we plan to tackle this specific topic in a brief communication that will be soon submitted to TCD (Kittel *et al.,* in preparation) rather than including this in a supplementary section of the current paper.

To address the reviewer's comment about uncertainties of our results, we propose to present in supplementary materials the following 2D comparison between our 50 km reference simulations (MAR50 hereafter) and the Agosta *et al.* (2018)'s 35 km results (MAR35 hereafter) and a map illustrating the biases of MAR forced by ERA-Interim at 50 km compared to the SMB observations.

[Figure]

[Figure]

a) 35km SMB (Agosta et al., 2018) (kg m⁻² yr⁻¹)

b) 50km SMB − 35km SMB (kg m⁻² yr⁻¹)

*Figure 1. a: Mean SMB simulated by MAR over 1979 – 2015 from Agosta et al.(2018). b: Comparison between the MAR SMB at a 35 km resolution from Agosta et al. (2018) and the MAR SMB at a 50 km resolution (this study). Units are kg m⁻² yr⁻¹. Non-significant anomalies (i.e., lower than the interannual variability) are hatched.*

[Figure]

MAR biases (kg m-2 yr-1)

*Figure 2. Comparison between MAR SMB and observed SMB from the GLACIOCLIM-SAMBA database (Favier et al., 2013) for 1950 – 2015. Units are kg m⁻² yr⁻¹.*

Figure 1 illustrates the SMB anomaly between MAR50 and MAR35. The largest anomalies are found over the Antarctic Peninsula and areas with a high orography spatial variability such as the Transantarctic Mountains. The significant anomalies are however smaller than the significant SMB anomalies due to changes in SSC from our study (except for the nearest pixels to the ocean where the MAR50 and MAR35 ice masks differ). Furthermore, they are mainly smaller than the SMB anomalies between MAR35 and RACMO2 presented in Agosta et al. (2018). Finally, it should also be noted that MAR50 biases compared to the observations are smaller than the SMB anomalies due to SSC changes (Fig. 2).

Besides adding Figure 1 and Figure 2 in supplementary materials, we also propose to modify the Section 3 "Evaluation against SMB observations" with a spatial analysis of MAR50 biases compared to the SMB observations as follows:

P7L11-18 *The high value of the correlation coefficient (r=0.93) between observed and modelled SMB values shows that MAR correctly represents the Antarctic SMB spatial variability at 50 km resolution over the 1979 – 2015 period (Fig. 2). Except over the Dronning Maud Land, the margins of the Amery ice shelf and a transect in the Wilkes Land, MAR overestimates the SMB (locally up to a factor of 5, Fig. S2).*

*As also shown by Franco et al. (2012) over the Greenland ice sheet, these biases could partially arise from the coarse resolution used here (50 km) which induces a topography smoothed at the ice sheet margins. This leads to an*

*unsatisfactory representation of the topographic barrier effect allowing the precipitation systems modelled by MAR to penetrate too far inland.*

*In order to estimate the biases and the uncertainty related to our resolution, the reference SMB of this study was briefly compared to the SMB at 35 km resolution from Agosta et al. (2018) (Fig. S3). This comparison also shows an SMB overestimation in the reference run compared to SMB results at a higher resolution, although this overestimation appears to be non-significant. The largest anomalies can be found over the Antarctic Peninsula and areas with a high orography spatial variability such as the Transantarctic Mountains. The coarse 50 km resolution used here leads to artificially overestimated precipitation on the windward sides of orographic barriers. This is notably the case over the Filchner-Ronne and Ross ice shelves where the Amundsen Sea Low generates a return flow. However it should be noted that the SMB anomalies of our 50 km reference run compared to both 35 km results (Agosta et al. 2018) and observations are smaller than the SMB anomalies due to SSC perturbations presented in Section 4.*

4. I would strongly advise to replace Fig. 3 by Fig. S2 in the main manuscript, as it displays inefficient to move back and forth to the supplementary material to visualize and compare results from experiments not shown in the main manuscript.

Thank you for the advice, we will replace Fig. 3 by Fig. S2.

**Substantive Comments**

**Section 2.3.4: P5 L1-2:** Here, I understand that monthly SSC (1979--2005) derived from CMIP5 ensemble average and two extreme members, as well as from ERA--Interim are interpolated to the MAR grid (50x50 km) using an inverse distance weighting method based on the four CMIP5 models/ERA--Interim cells nearest to the current MAR one. If so, please reformulate accordingly.

**L5:** This is confusing, are the authors calculating monthly mean SIC--SST from CMIP5/ERA--Interim for 1979--2005 (12 values) or an annual mean (1 value). I understand that monthly SIC and SST anomalies are used, please clarify.

We calculated monthly mean SIC and SST anomalies between CMIP5 and ERA-Interim so as to take into account open water areas when computing the monthly SST anomalies and to not introduce additional temperature biases: monthly SST anomalies were computed only if SIC from both CMIP5 ensemble average and ERA-Interim are less than 50%. Monthly values of SIC and SST anomalies are then averaged to obtain an annual anomaly value, supposed to represent a constant bias.

We suggest to reformulate P5 L1-5:

*For that purpose, we have determined a perturbation whose magnitude is representative of the CMIP5 ensemble bias. Monthly SSC over 1979–2005 from all the CMIP5 models (using the historical scenario), as well as from ERA-Interim were interpolated on the MAR grid (50 km x50 km) using an inverse-distance weighted method based on the four CMIP5 models/ERA-Interim grid cells nearest to the current MAR one. We then computed the CMIP5 ensemble average from the interpolated CMIP5 monthly SSC. Firstly, to not introduce additional temperature biases, monthly SST anomalies between CMIP5 and ERA-Interim were computed only if the SIC from both the CMIP5 ensemble average and ERA-Interim are less than 50%. Secondly, we average the monthly anomalies to obtain a mean anomaly, supposed to represent a constant bias over time.*

**L6--10:** Are the new 6--hourly SST--SIC calculated as the sum of 6--hourly ERA-- Interim (i.e. for a specific day of a certain month) and the corresponding monthly anomaly in SST--SIC from CMIP5 models? If so, please reformulate.

Yes, it is. We propose to reformulate P5l6-10 as follows:

*New 6-hourly forcing SST are calculated as the sum of the 6 hourly ERA-Interim (i.e., for a specific day of a certain month) and the corresponding monthly anomaly in SST from CMIP5 ensemble average (Fig. 1b), hereafter referred to as SST(CMIP5) experiment. In the same way, we define SIC(CMIP5) experiments in which SIC anomalies (Fig. 1e) from the CMIP5 ensemble average are added to the 6-hourly ERA-Interim SIC.*

**P7 L17--18:** This is an important caveat that should be addressed in an additional section "Limitations". A 2D map comparison between 50 km and 35 km simulations could help the reader understanding where large uncertainties are likely to be found at 50 km, see also general comments #1 and #3.
See our answer to General comments 1-3.

**Point Comments**

**P1 L10:** The authors refer to "warm SSC" or "cold SSC" several times across the manuscript. While it may sound obvious that warm (resp. cold) SSC represents combined low SIC and high SST (resp. high SIC and low SST), this should be explicitly stated in the manuscript, e.g. in Section 2.3.3 or more generally in Section 2.3.

Warm and cold SSC are explicitly stated in the manuscript P14,L5 (Section Conclusion). We will define warm (i.e., high SST and low SIC) and cold (i.e., low SST and high SIC) SSC by modifying the abstract (P1,L10) and the first reference to "warm" or "cold " SSC (P6,L5) according to

P1,L10 *Results show increased (resp. decreased) precipitation due to perturbations inducing warmer, i.e. higher SST and lower SIC (resp. colder, i.e. lower SST and higher SIC) SSC than ERA-Interim significantly altering the SMB of coastal areas, as precipitation is mainly related to cyclones that do not penetrate far into the continent*

And

P6,L5 *Following the same method, we perform combined experiments for two selected CMIP5 models, namely NorESM1-ME (Bentsen et al., 2012) and GISS-E2-H (Schmidt et al., 2014), respectively representative of a colder (i.e., lower SST and higher SIC) and warmer (i.e, higher SST and lower SIC) SSC than ERA-Interim as shown in (Agosta et al., 2015).*

**P2 L29--30:** What do the authors mean by "neither feedbacks involving sea ice and ocean"?
We suggest to delete this part of the sentence as it was redundant with the beginning *"This means that we do not consider feedbacks on the general circulation associated to sea ice removal (e.g., Bromwich et al., 1998; Krinner et al., 2014)"*

**P3 L18:** Could the authors briefly elaborate on the reasons why the drifting snow module is switched off?
Similarly to Agosta et al. (2018), we decided to switch off the drifting snow as the new version of this module is still under evaluation against satellite and ground-based observations over the whole Antarctic ice sheet. We suggest to add this reason in our manuscript P3L1*8:*

*Although MAR includes a drifting snow module (Gallée et al., 2001), this module has been switched off similarly to Agosta et al. (2018) as the new version of this module is still under evaluation against satellite and ground-based observations.*

**L25:** Could the authors define OSTIA?
The Operational SST and Sea Ice Analysis (OSTIA) is a daily global SST analysis produced at a 0.05° resolution (Stark *et al.,* 2007; Donlon *et al*., 2012). We suggest to change the text P3L25 as follows:

*It is worth noting that ERA-Interim uses the SST and SIC values from ERA-40, which are based on monthly and weekly ocean forcing fields (Fiorino, 2004), until January 2002. Afterwards, a switch was made with the daily operational NCEP product and since 2009 with the Operational SST and Sea Ice Analysis (OSTIA). The latter is a daily global SST analysis product at a 0.05° resolution (Stark et al., 2007; Donlon et al., 2012)*

**L31:** I guess the authors mean "[…] from a previous reference simulation", or is the initialization based on multiple simulations, please clarify.
Indeed, we use a snowpack from a previous reference simulation. Thank you for the correction.

**P4 L2:** Could the authors mention the original resolution of their DEM?
The original resolution of the DEM is 1km. We add it P4L2:
*The Antarctic topography is based on the 1-km resolution DEM Bedmap2 from Fretwell et al. (2013).*

**P5 L7:** I guess the authors mean (Fig. 1b). **L8:** (Fig. 1e).

Yes, thank you. (See our answer to the first point comment where we already corrected it).

**P6 L3:** Could the authors estimate by how many ºC on average these two extreme CMIP5 members are "colder" or "warmer" than ERA--Interim.
The mean temperature anomaly as well as the mean SIC anomaly is listed in Table 1. We will specify it in our manuscript P6L3:

*Table 1 compares SSC perturbations to the reference SSC for June-July-August (JJA) and December-January-February (DJF) SST and sea ice area (SIA). The mean SST and SIC anomalies of CMIP5 ensemble average, NorESM1-ME, and GISS-E2-H are also listed in Table 1.*

**L4--5:** I guess the authors mean: (Fig. 1a,d) and (Fig. 1c,f.)
Yes, Thanks. We modify P6L4-5 accordingly:
*These experiments are hereafter called SST/SIC(NorESM1-ME) (Fig. 1a,d) and 5 SST/SIC(GISS-E2-H) (Fig. 1c,f).*

**P7 L4:** Could the authors mention how many measurements were discarded from the evaluation?
The original SMB data base from Favier *et al.* (2013) contains 3236 observations among which, as indicated p7,L3, 206 observations do not fit our selection criterions (i.e, covering more than 8 years if the observations interval is not included in the period 1979-2015).
**P9 L2-7:** Modeling limitations in the AP could be discussed in an additional "Limitations" section, see general comment #3.
As we have dedicated a full paper to the influence of the resolutions and strengthened the discussion on the biases related to resolution (see answer to General comments 1-3), we think that an additional "Limitations" section for discussing only the limitations over the AP is not necessary anymore.

**P10 L11--12:** Do the authors mean that the SMB from SIC(CMIP5) does not significantly differ from the reference, both spatially and integrated over the whole AIS? If so, please reformulate.
Yes, it is. We reformulate P10-L11-12:
*Finally, the mean SMB from SIC(CMIP5) does not significantly differ from the reference SMB, both spatially (Fig. 3j) and integrated over the whole AIS (Table 2.)*

**L16:** "SST+2/SIC--3", same at **L27**.
Thanks.
**L19--20:** I do not see snowfall decreasing over the AP in the supplementary figures, this rather seems to occur in the surrounding ocean. Please clarify.
The significant decrease in snowfall that we described "over the AP" only occurs over the Larsen C and George VI ice shelves. We will clarify it by changing P10 L19-20:

*Moreover, snowfall significantly decreases over Larsen C and George VI ice shelves (both located in the AP) but is largely compensated by rainfall refreezing into the snowpak.*

**L32:** The authors certainly mean (Fig. 1f).
Yes, corrected thank you.

**L32--33:** "SST/SIC(CMIP5) suggests", what do the authors mean by "as SIC and SST anomalies […] around the mean"? Please, clarify.
We mean that the CMIP5 average anomalies for both SIC and SST are weak in our experiment as CMIP5 models are more or less equally distributed (warm or cold SSC anomalies) around the ERA-Interim SSC, even if the mean CMIP5 SSC are slightly warmer (lower SIC and higher SST) than ERA-Interim (see Table 1). We suggest to modify P10 L32-33 by:

*SST/SIC(CMIP5) suggests a non-significant positive anomaly for both integrated and spatial SMB (Fig. 3k) as the mean SIC and SST anomalies in CMIP5 models do not significantly differ to the ERA-Interim SSC (Table 1, Fig. 1b,e). CMIP5 models anomalies are more or less equally distributed (warm or cold SSC anomalies) around the ERA-Interim*

*SSC, even if the mean CMIP5 SSC are slightly warmer than ERA-Interim explaining the non-significant positive SMB anomaly.*

**P11 L7--9:** I am not sure to understand the links between unchanged inland temperature, coastal precipitation enhancement and downward (katabatic?) winds. Could the authors reformulate this sentence?
We suggest to reformulate P11 L7-15 as follows
*These results suggest that precipitation can be formed further inland depending on the properties of air masses. In agreement with Gallée (1996), our hypothesis is that colder and drier air masses in cold ocean experiments are not sufficiently loaded with moisture to enable saturation and then snowfall over the margins. The decrease in moisture is likely to be larger than the decrease in the maximal moisture content in the atmosphere associated to lower temperatures. This leads to a larger amount of remaining humidity that can be advected further inland (Fig. 4b,d and S10b,d) where saturation occurs because of the lower temperatures. On the opposite, the additional humidity in warm ocean experiments results in air masses that reach saturation faster (the increase in humidity overcompensates the increase in the maximal moisture content) and thus generating precipitation over the ice sheet slopes. MAR also simulates significantly higher upper air temperature over the central part of the ice sheet (Fig. S11c,e and Fig S12c,e) that, combined with the lower remaining humidity, (Fig. 4C,e and S10c,e) limit snowfall. (see also our response to R#2)*
**L10-13:** This discussion should be moved to a new section "Limitations".
We suggest to move this discussion in the section "Results" (see our answer to general comments 1-3)

**L29--31:** Please reformulate this sentence "present--climate […] 21st century", it reads better at **L18--19 of P14**.
We will reformulate P11 L29-31 by:

[…]Our sensitivity tests with warmer (CMIP5-based) oceans reveal SMB anomalies over the current climate in the lower range of the SMB increase projected for the end of the 21st century

**Stylistic suggestions**
Thank you for all the stylistic suggestions. We will take them into account in the revised version of our manuscript.

**P1 L2:** […] boundary forcing fields prescribed by reanalyses […]. **L7:** Altering is rather negative, and suggests that data have been deteriorated. I would suggest: "by modifying the ERA--Interim SSC". **L10:** Replace "altering" by "affecting". **L17:** "exchange of gas". **L18:** Remove "behaviour"; "impacts" or "affects" instead of "alters". **L19:** "[…] water vapour loading of air masses, potentially […]". **L20:** Remove "as the Antarctic" and insert "that" after "(SMB)". **L23:** The authors could consider reformulating as follows: "[…] have experienced a significant increase since the 1970s (e.g. Massonnet et al., 2013), highly contrasting with the dramatic decline reported in the Arctic Ocean […]".
**P2 L9:** Maybe "[…] and increased precipitation […]". **L13:** I suggest: "[…] (e.g. Weatherly, 2014), with simplified physics resulting […]". **L14:** I suggest: "[…] (RCMs) forced by former and less reliable reanalyses (e.g. ERA--15 in Bromwich et al., 2007) over short periods […]". **L24:** "[…] with the 'Modèle Atmosphérique Régional' (MAR) for the period 1979--2015. This allows partitioning […]". **L27--31:** I suggest: "[…], this study only discusses the direct […]. This means that no feedback on the […] removal     is considered […] Krinner et al. 2014). Only direct impacts on […] components are accounted for. Note    that the general […]". **L34:** Replace "altered" by "perturbed".
**P3 L7:** Maybe "The effect of sea spray on […]". **L10:** I suggest: "[…] sub--modules, that simulate energy and […]". **L15:** Remove "Consequently,". **L16:** I suggest: "[…] and open--ocean fractions". **L21:** "forced by". **L27:** "[…] evolve as a function of accumulated snowfall and surface […]". **L29:** "we start". **L33:** "[…] moisture source for precipitation over the AIS […]".
**P4 L1:** Maybe "selected" instead of "chosen". **L2:** "extending 6 km above". **L5:** "[…] SMB to SSC perturbations is limited to the […] SIC anomalies within the MAR […]". **L9--10:** "perturbed" instead of "modified". **L13--16:** "for ice--free pixels", "converted into full ice--covered pixels if the SST drops […] (-- 2ºC). For an SST […] SIC value is set to […]". **L18:** Maybe "at the interface between […]". **L22:** "[…] and maintain SST of sea ice--covered pixels […]". **L25:** "in sea ice extent associated". **L29:** Replace "present climate" by "present--day" or "contemporary".
**P5 L9:** "[…] anomalies into the original […] enables to account for constant […]".
**P6 L9:** "perturbed" instead of "altered". **L10:** I suggest: "SST and/or SIA anomalies for JJA and DJF periods are 1.5 times as large as CMIP5 mean […]". **L14:** "brief" instead of "short". **L14:** This is rather negative, I would suggest: "to

highlight the impact of using a coarser horizontal resolution on SMB representation". **L15:** Remove "the one described".

**P7 L1:** Replace "by" by "using". **L2:** "observations collected prior to our study period". **L3:** "for the same period.". **L6:** Add "the" before "observation locations". **L7:** I suggest: "(Agosta et al., 2012), unresolved at 50 km […]". **L8:** Replace "consequently" by "i.e." or by "or". **L13:** "noted" instead of "noticed". **L16:** "significantly smoothed topography".

**P9 L1:** "enhanced" instead of "stronger". **L7:** "This allows locating". **L12:** For consistency, "interannual variability". **L14:** "The warmer ocean leads to […] of similar magnitude as […] converting snowfall into rainfall." **L16:** "[…] higher temperature also causes […] SST+4 relative to the reference simulation (Fig. S7a--l)." **L18--20:** "[…] sublimation are larger in SST+4 (Fig. S6a) because […] temperature, increased precipitation dominates and the SMB anomaly is significantly positive (Fig. 3a and Table 2)." **L22:** "ocean surface and reduce the water vapour". **L26:** "Over the plateau, larger deposition combined with snowfall […] than for the reference run.". **L28--29:** "[…] colder ocean as lower SST also decrease the near--surface air temperature […]". **L31:** "revealing similar patterns as in SST+2 (Fig. S2m) although non--significant […]".

**P10 L1:** "smaller" instead of "lower". **L5--7:** "[…] decrease in precipitation is observed over the new ice--covered ocean […] act as an insulator […] 2--m air temperature by 10ºC". **L9:** Maybe "pronounced" instead of "strong"; "[…] SIC--6/SIC+6). This is likely due to the smaller magnitude […] SIC--3 compared to the magnitude of SIC extension […]". **L23:** "show similar anomalies as". **L24:** "larger anomalies […] Besides, the sensitivity of AIS SMB to SSC is non--linear".

**P11 L2:** Maybe "marginal" instead of "external". **L14--15:** I suggest: "opposite pattern: drier air masses have to rise up higher […] so that precipitation is generated further inland.". **L22:** "affect" instead of "modify". **L23:** "three times as large". **L29:** "perturbed SSC".

**P13 L1:** "are by far smaller". **L2:** Maybe "showing" instead of "with potential"; the authors could consider "[…] SMB anomalies do not significantly differ from observed SMB". **L4:** Replace "assess" by "evaluate". **L7:** Replace "driven" by "forced".

**P14 L2:** Replace "altering" by "perturbing" and move "unchanged" to the end of the sentence. **L3:** "The first set consists of […]". **L8:** "(resp. decreased) precipitation due to warmer (resp. colder) […]". **L12:** Maybe "leads to earlier/faster saturation as they rise […]". **L15:** I suggest: "However, comparing modelled SMB from sensitivity experiments with observations shows no significant difference, suggesting […]". **L18:** "with warmer (CMIP5--based) SSC reveal that SMB […] climate stand in the lower". **L20:** Remove "as demonstrated in this study". **L21:** "using potentially biased SSC as forcing". **L22:** Replace "produce new" by "carry out future".

**Figures and Tables**

**Table 1:** For the reference run, insert "0.00" or "--" to fill the blanks in the anomaly columns.
Ok, we will fill the blanks with "-" in our revised manuscript.
**Table 2:** Caption **L2:** "floating ice" instead of "not grounded ice".
We will modify accordingly the caption of Table2.
For comparison with the coarser resolution simulation, the authors should consider adding integrated numbers from the 35 km run discussed in Agosta et al. (2018). See also general comment #2.
See our response to general comment 1-3
**Fig. 2:** For comparison, the authors should consider including a second scatterplot showing outputs of the 35 km run.
See our response to general comment 1-3
**Fig. 3:** This figure could be replaced by Fig. S2. See also general comment #4.
Thank you for the suggestion, Fig 3 will be replaced by Fig S2
**Fig. S6:** Caption **L1:** "minus" instead of "mines".
Thank you for the correction

We would like to thank the reviewer #2 for his constructive comments which will help to improve our manuscript.

1. You perturb both the SST and the SIC, but not necessarily in a consistent way. In my opinion, more material should be given to illustrate a consist perturbation, for example by comparing how the SST bias in the GCM compares to the SIC bias? Another way would be to assume – for example – that meridional SST gradient remains unchanged as the SST increases, which imposes the retreat of the sea-ice edge.

We recognize that our method does not necessarily lead to consistent SIC perturbations associated to the SST perturbations since SIC perturbations depend on the number of neighbouring pixels taken into account. However, the experiments has been designed in order to study the effect of perturbations similar to SST and SIC biases in the GCMs (see Table 1 and p6, L9-11). Our SIC and SST perturbations can also be compared to previous works such as Van Lipzig et al. (2002). They reduced the mean sea ice cover by 50% for a 2°C temperature rise (values also suggested by Thompson and Pollard (1995) in a 2°C warmer climate), which is close to our SIC perturbations (-53% in winter).

Furthermore, the methods used in this study lead to smoothed SST field preventing abrupt SST changes near the sea ice edge, and also enable to modify polynya extents as SIC can vary from 0 to 100% in MAR. This would not have been possible with a method using a retreat of sea-ice edge only based on an unchanged meridional SST gradient. We would also encounter difficulties for determining the sea-ice edge as the MAR ice mask is not a binary mask. Assuming that the meridional SST gradient remained unchanged is also a strong hypothesis that still have to be demonstrated as the meridional SST gradient strongly depends on the presence/absence of the sea ice. In the context of the polar amplification (even less strong in the southern hemisphere), the gradient can be expected to decrease but if some sea ice remains in a warmer climate, it would constrain the SST at the freezing point over the highest latitudes while increasing at lower latitudes. They are thus two opposite effects, probably leading to high uncertainties about the meridional SST gradient (François Massonnet, UCL, personal communication, 2018). Although it is an interesting debate and research, we think that the future of the meridional SST gradient is beyond the scope of this study and we have thus preferred not to rely on the hypothesis of an unchanged meridional gradient.

The aim of using CMIP5 anomalies was also to apply to the ERA-Interim SSC a SIC perturbation related to the SST bias. However, it appears that SIC biases in CMIP5 models are not only related to SST but also to process parameterization such as lateral melting (e.g., Roach et al., 2018) so that SST biases and SIC biases could not be consistently derived from one to another.

For all those advantages compared to disadvantages of each method, we have preferred to follow the methods defined by Noel et al. (2014) for constructing our SSC perturbations. We suggest to clearly report that our sensitivity experiments does not necessarily lead to consistent SIC perturbations associated to SST changes by adding this discussion to P6L6.

P6, L6: *Table 1 compares SSC perturbations to the reference SSC for June-July-August (JJA) and December-January-February (DJF) SST and sea ice area (SIA). The SIA is defined as the sum of the products of the SIC and area of all grid cells with a SIC value of at least 15%. SIA is preferred to sea ice extent because it better accounts for SIC variations (Roach et al., 2018). Sensitivity experiments with altered SST by ±2 °C and SIC with the ±3 neighbour pixels are in the range of CMIP5 anomalies. Other perturbations (SST±4 and SIC±6) represent a 1.5 times larger anomaly in SIA and/or SST for both JJA and DJF mean values than CMIP5 mean anomalies over the current climate. However, it should be remembered that our sensitivity experiments are not based on climatological consistent SIC (resp. SST) perturbations related to SST (resp. SIC) perturbations. For instance, the SIC prescribed in our experiments associated to 2°C warmer SST could be significantly different from the real SIC in a 2°C warmer climate since we do not use SIC projections from a GCM.*

2. I do not really following the reasoning throughout the paper why there is more precipitation inland when SST is lower or SIC is higher. You argue that this is because the dryer air has to rise up higher to reach saturation. Although, this is of course true, it does not imply that precipitation can be brought higher up – it just means the saturation point is at a higher elevation. For saturated air, the amount of moisture transported in the interior is only dependent on temperature and circulation. So additional analyses are needed to shed light on this issue. The best way would be to do a moisture budget over the interior and see whether small circulation changes might be responsible for this.

Although you do spectral nudging, circulation close to the surface might deviate which can be relevant for moisture advection. Although this comment is valid for the entire results/discussion section, p11, line 14/15 is particularly misleading.

We do not mean that precipitation can be brought higher up, but only that the saturation is likely to occur at a higher elevation. In cases of marine air (i.e., with a high humidity content) intrusion towards the central part of the ice sheet, precipitation is then formed further inland (p11, L13-15).

The indiscriminate nudging applied to the upper atmosphere of the model is designed to prevent any wind deviation from the forcing. Figure 1 presents the mean near-surface (2 m) wind speed in the reference simulation and wind anomalies for both wind speed and direction in SST-4/SIC/6 and SST+4/SIC-6 experiments. It highlights the absence of a wind deviation but shows a strengthening (resp. weakening) of the flow in a warmer (resp. colder) ocean over both the ice sheet and the ocean. However, these changes are mainly significant over areas where sea ice is removed or added similarly to Gallée (1996) and Van Lipzig *et al.*(2002). These changes are due on one hand to the surface roughness strongly modified over the ocean. On the other hand, the higher (resp. lower) temperature contrast between the ocean and the atmosphere reinforces (resp. weakens) the ice-breeze effect as described by Gallé. (1996). Although we show here the mean surface flow and direction over 1979-2015, it is also true for any specific day.

[Figure]

a) Mean near-surface wind speed        b) SST-4/SIC+6        c) SST+4/SIC-6

*Figure 1. a: Mean near-surface (2m) wind speed modelled by MAR over 1979 – 2015. Difference in mean surface wind speed (m/s) between the reference simulation and (b) SST-4/sSIC+6, (c) SST+4/SIC-6 experiments. Wind speed differences lower than the interannual variability are considered as non-significant and are dashed. The wind direction is also indicated with black (resp. green) vectors for the reference simulation (resp. sensitivity experiments).*

The large amount of MAR simulations did not enable us to store the atmospheric variables at each vertical level of the model preventing us to compute a moisture budget over the ice sheet. As an alternative, we propose to analyze the specific humidity (Figs 2 and 3) and temperature (Figs 4 and 5) at 600 hPa (Figs 2a,b,c,d,e and 4a,b,c,d,e) and at 700 hPa (Figs 3a,b,c,d,e and 5a,b,c,d,e). We found a negative anomaly at the ice sheet margins but a higher specific humidity over the central part of the ice sheet in SST-4/SIC+6 (Fig 3b and Fig 4b). On the opposite, the specific humidity is significantly increased in the SST+4/SIC-6 over the margins and decreased over the central region (Fig 3c and Fig 4c). We also compared the snowflakes content between the sensitivity experiments (SST-4/SIC/6; SST+4/SIC-6) and our reference simulation (not shown). These anomalies are very similar to the snowfall anomalies pattern presented in our supplementary materials (Figure 4, p5) with for instance, higher snowflake concentration (up to 30%) over the central ice sheet in the SST-4/SIC+6 experiment and lower snowflake concentration over the same area in SST+4/SIC-6 (up to -30%).

These results suggest that precipitation can be formed further inland depending on the properties of air masses. In agreement with Gallée (1996), our hypothesis is that colder and drier air masses in cold ocean experiments are not sufficiently loaded with moisture to enable saturation and then snowfall over the margins. The lack in moisture is likely to overcompensate the lower temperature. This leads to a larger amount of remaining humidity that can be advected further inland (Figure 2b and 3b) before saturation occurs due to lower temperatures. On the opposite, the additional humidity in warm ocean experiments results in air masses that reach saturation faster (humidity still overcompensates the higher temperature) and thus generating precipitation over the ice sheet slopes. MAR also simulates significantly higher air temperatures over the central part of the ice sheet (Figure 4c and 5c) that, combined with the lower remaining humidity, (Figure 2c and 3c) limit snowfall.

[Figure]

Figure 2. a: Mean specific humidity modelled by MAR over 1979–2015 at 600 hPa (Units: g/kg). Difference in mean specific humidity (%) between the reference simulation and (b) SST-4/SIC+6, (c) SST+4/SIC-6, (d) SST-2/SIC+3, (e) SST+2/SIC-3 experiments. Differences lower than the interannual variability are considered as non-significant and are dashed.

[Figure]

Figure 3. Idem as Figure 2 but at 700 hPa.

[Figure]

Figure 4. Idem as Figure 2 but for the mean temperature (°C) at 600 hPa.

[Figure]

Figure 5. Idem as Figure 2 but for the mean temperature (°C) at 700 hPa.

We suggest to clarify our explanation and add the Figure 2 in our manuscript, Figures (3-5) in supplementary materials and modify P11 L7-15 by

*These results suggest that precipitation can be formed further inland depending on the properties of air masses. In agreement with Gallée (1996), our hypothesis is that colder and drier air masses in cold ocean experiments are not sufficiently loaded with moisture to enable saturation and then snowfall over the margins. The decrease in moisture is likely to be larger than the decrease in the maximal moisture content in the atmosphere associated to lower temperatures. This leads to a larger amount of remaining humidity that can be advected further inland (Fig. 4b,d*

*and S10b,d) where saturation occurs because of the lower temperatures. On the opposite, the additional humidity in warm ocean experiments results in air masses that reach saturation faster (the increase in humidity overcompensates the increase in the maximal moisture content) and thus generating precipitation over the ice sheet slopes. MAR also simulates significantly higher upper air temperature over the central part of the ice sheet (Fig. S11c,e and Fig S12c,e) that, combined with the lower remaining humidity, (Fig. 4C,e and S10c,e) limit snowfall.*

3.On p 11 line 19 you state that 'Katabatic winds prevent significant impacts of SSC on the Antarctic SMB'. Although this might be true, I do not see proof for this in the manuscript. Even if there would be no katabatics, the fact that air has to rise over the topographic barrier and additional moisture is constrained to the boundary layer, might be enough to prevent significant effect.

Similarly to Gallée (1996) and Noel et al. (2014), we found a strengthening of the near-surface katabatic flow associated to lower SIC and higher SST (Figure 6c). Increased katabatic winds bring more cold air masses from the central ice sheet and cool the ice sheet margins. Furthermore, they also export humidity away from the continent (Van Lipzig *et al.*, 2002). However, this effect is limited to the katabatic layer.

Our deepest analysis about the humidity and temperature suggests that a significant part of the additional moisture is not constrained to the boundary layer and reaches upper atmospheric layers (600 hPa or ~4km height) for the experiments with the strongest SSC perturbations (Fig 2b,c and 3b,c). This contrasts with the results presented in Van Lipzig *et al.* (2002) where the surface anomalies were restricted below the lowest 1-2km. The blocking effect due to the topographic barrier is likely to be reduced as these large anomalies reach higher atmospheric levels, contrary to experiments with slightly perturbed SSC (SST+-2/SIC-+3) where anomalies remain confined in the low levels.

[Figure]

a) Mean surface wind (m/s)     b) SST−4/SIC+6 (m/s)     c) SST+4/SIC−6 (m/s)

Figure 6. Idem as Figure 1 but without wind vectors and only on the ice sheet.

We thus propose to clarify of our statement P11 L19-21 about the effect of katabatic winds as well as our refutation about the topographic barrier and the additional moisture. We also suggest to add Figure 6 in supplementary material

*Similarly to Van Lipzig et al.(2002), moisture and temperature anomalies remain confined below 700 hPa in the experiments with slightly perturbed SSC (SST+-2/SIC-+3) (Fig S10d,e and Fig S12d,e). On the opposite, in the experiments with the largest SSC perturbations (SST+-4/SIC-+6), a significant part of the additional moisture is not constrained to the boundary layer and reaches upper atmospheric layers (600 hPa) (Fig 4b,c). The blocking effect due to the topographic barrier is likely to be reduced suggesting that these large anomalies can have a deeper influence inland.*

*Furthermore, katabatic winds are enhanced when the SIC is decreased and the SST increased (Fig S13c) as already shown in Gallée, 1996; Van Lipzig et al., 2002. Due to their offshore direction, they prevent the influence of warm ocean anomalies by precluding their propagation at the surface of the ice sheet and by advecting cold air from inland regions towards the margins.*

Minor comments:

1. Abstract: last sentence: a number for a sensitivity in % is meaningless when the magnitude of the perturbation is not specified. Please clarify in the abstract

We think that giving a magnitude of the perturbation is meaningless as the SSC in these experiments are computed with the CMIP5 biases that significantly differ spatially. We therefore propose to specify that the SSC perturbations are based on the CMIP5 biases in the sentence of *P1 L13*

*Sensitivity experiments with warmer SSC based on the CMIP5 biases reveal integrated SMB anomalies (+5% – +13%) over the present climate (1979 – 2015) in the lower range of the SMB increase projected for the end of the 21st century*

2. P1, line 22: I am not sure if I follow the definition of the Sea Ice Extent given there. Can you give a reference for this definition or clarify?

This definition can be notably found in Parkinson and Cavalieri (2012), Cavalieri and Parkinson (2012), Roach *et al. (*2018) (all cited in our manuscript) as well as in Vaughan *et al.* (2013).

We propose to slightly modify the definition to use the exact same definition:

*P1l22: generally defined as the area of all grid cells of satellite or model products with a SIC of at least 15%*

3. P 2, l11: reference is van Lipzig et al., (2002) not van Lipzig and van Meijgaard (see below). Van Lipzig, N.P.M., E. van Meijgaard and J. Oerlemans, 2002. Temperature sensitivity of the Antarctic surface mass balance in a regional atmospheric climate model. J. Clim., 15(19), 2758-2774. doi:10.1175/1520-0442(2002)0152.0.CO;2.

Thank you for the correction of the reference. We have also identified a second reference mistake P2,L2. Turner *et al.* (2013) was right but the interesting paper for our study is:

Turner, J., Bracegirdle, T. J., Phillips, T., Marshall, G. J., & Scott Hosking, J. An initial assessment of Antarctic sea ice extent in the CMIP5 models. *Journal of Climate*, 26(5), 1473–1484. https://doi.org/10.1175/JCLI-D-12-00068.1, 2013

Both references will be corrected in the revised version of our manuscript.

4. P9: l12: Air does not have 'a capacity' to hold water vapour. The water vapour is one of the components of air. Please reformulate.

Ok, we suggest to modify *P9 L12* by:

*The higher evaporation and inherent increase in air moisture content [..]*

b) 50km SMB − 35km SMB (kg m$^{-2}$ yr$^{-1}$)

**Figure S3.** a: Mean SMB simulated by MAR over 1979 – 2015 from Agosta et al.(2018). b: Comparison between the MAR SMB at a  35 km resolution from Agosta et al. (2018)  and the MAR SMB at a 50 km resolution (this study) Units are kg m $^{-2}$ yr$^{-1}$. Non-significant anomalies i. e., lower than the interannual variability) are hatched.

[Figure]

**Figure S4.** Difference in mean annual total precipitation (rainfall + snowfall) (kg m$^{-2}$ y$^{-1}$) between the reference simulation and (a) SST+4, (b) SIC-6, (c) SST+4/SIC-6, (d) SST/SIC(GISS-E2-H), (e) SST-4, (f) SIC+6, (g) SST-4/SIC+6, (h) SST/SIC(NorESM1-ME), (i) SST(CMIP5), (j) SIC(CMIP5), (k) SST/SIC(CMIP5), (m) SST+2, (n) SIC-3, (o) SST+2/SIC-3, (p) SST-2, (q) SIC+3, (r) SST-2/SIC+2 experiments. Difference less than the interannual variability are considered as non-significant and are dashed. l) Mean annual SMB (kg m$^{-2}$ y$^{-1}$) simulated by MAR forced by ERA-Interim over 1979 − 2015.

[Figure]

**Figure S5.** Same as Fig.  S4 but for snowfall over the ice sheet and the surroudning ocean.

[Figure]

**Figure S6.** Same as Fig.  S4 but for rainfall over the ice sheet and the surroudning ocean.  White areas over the ice sheet indicates that there is no rainfall.

[Figure]

**Figure S7.** Same as Fig. ?? S4 but for waterfluxes (sublimation mines deposition) at the ice sheet surface. Positive fluxes indicates sublimation while negative fluxes are representative of deposition processes.

[Figure]

**Figure S8.** Same as Fig.  S4 but for meltwater production at the surface.  White areas over the ice sheet indicates that melt never occurs.

[Figure]

**Figure S9.** The Antarctic ice sheet and surrouding seas. Elevation contours are shown every 1000 m.

[Figure]

**Figure S10.** a: Mean specific humidity modelled by MAR over 1979–2015 at 700 hPa (Units: g/kg). Difference in mean specific humidity (%) between the reference simulation and (b) SST-4/SIC+6, (c) SST+4/SIC-6, (d) SST-2/SIC+3, (e) SST+2/SIC-3 experiments. Differences lower than the interannual variability are considered as non-significant and are dashed.

[Figure]

**Figure S11.** a: Mean air temperature modelled by MAR over 1979–2015 at 600 hPa (Units: °C). Difference in mean air temperature (°C) between the reference simulation and (b) SST-4/SIC+6, (c) SST+4/SIC-6, (d) SST-2/SIC+3, (e) SST+2/SIC-3 experiments. Differences lower than the interannual variability are considered as non-significant and are dashed.

[Figure]

a) Mean Temperature  b) SST−4/SIC+6  c) SST+4/SIC−6

d) SST−2/SIC+3  e) SST+2/SIC−3

**Figure S12.** Same as Fig. S11 but 700 hPa.

**Table S1.** Top: Annual mean integrated (Gt yr$^{-1}$) and standard deviation (Gt yr$^{-1}$) total precipitation (rainfall and snowfall), snowfall and rainfall ver the whole AIS (including grounded and not grounded ice) for the reference simulation (1979–2015). Bottom: Difference of annual mean total precipitation (rainfall and snowfall), snowfall and rainfall (Gt yr$^{-1}$ and %) between each sensitivity test and the reference simulation (1979–2015). Anomalies larger than the inter-annual variability are considered as significant and are displayed in bold.

| Mean (Gt y$^{-1}$) | Total precipitation | Snowfall | Rainfall |
|---|---|---|---|
| Reference | $2678 \pm 110$ | $2658 \pm 109$ | $20 \pm 3$ |
| Anomaly (Gt y$^{-1}$) | Total precipitation | Snowfall | Rainfall |
| SST-4 | -64 | -61 | **-3** |
| SST-2 | -89 | -85 | **-4** |
| SST+2 | +50 | +45 | **+5** |
| SST+4 | **+162** | **+137** | **+25** |
| SIC+6 | **-170** | **-166** | **-4** |
| SIC+3 | -107 | -104 | **-3** |
| SIC-3 | +25 | +28 | **-3** |
| SIC-6 | +91 | +93 | -2 |
| SST-4/SIC+6 | **-136** | **-133** | **-3** |
| SST-2/SIC+3 | **-129** | **-125** | **-4** |
| SST+2/SIC-3 | **+133** | **+126** | **+7** |
| SST+4/SIC-6 | **+344** | **+304** | **+40** |
| SST/SIC-Nor | -105 | -102 | **-3** |
| SIC-CMIP | +36 | +35 | +1 |
| SST-CMIP | +80 | +79 | +1 |
| SST/SIC-CMIP | +105 | +104 | +1 |
| SST/SIC-GIS | **+368** | **+353** | **+15** |

---

## Author Response (AR2)

Dear Editor,

We would like to thank you for your rereading and the corrections.

We found some additional typing errors or bad latex entries, corrected in our latest version.

Best regards,
Christoph K. on behalf of the co-authors

[revised manuscript text omitted]

**Figure S4.** Difference in mean annual total precipitation (rainfall + snowfall) (kg m$^{-2}$ yr$^{-1}$) between the reference simulation and (a) SST+4, (b) SIC-6, (c) SST+4/SIC-6, (d) SST/SIC(GISS-E2-H), (e) SST-4, (f) SIC+6, (g) SST-4/SIC+6, (h) SST/SIC(NorESM1-ME), (i) SST(CMIP5), (j) SIC(CMIP5), (k) SST/SIC(CMIP5), (m) SST+2, (n) SIC-3, (o) SST+2/SIC-3, (p) SST-2, (q) SIC+3, (r) SST-2/SIC+2 experiments.  Differences less than the interannual variability are considered as non-significant and are dashed. l) Mean annual SMB (kg m$^{-2}$ yr$^{-1}$) simulated by MAR forced by ERA-Interim over 1979 – 2015.

[Figure]

**Figure S5.** Same as Fig. S4 but for snowfall over the ice sheet and the  surrounding ocean.

[Figure]

**Figure S6.** Same as Fig. S4 but for rainfall over the ice sheet and the  surrounding ocean. White areas over the ice sheet  indicate that there is no rainfall.

[Figure]

**Figure S7.** Same as Fig. S4 but for  water fluxes (sublimation  minus deposition) at the ice sheet surface. Positive fluxes  indicate sublimation while negative fluxes are representative of deposition processes.

[Figure]

**Figure S8.** Same as Fig. S4 but for meltwater production at the surface. White areas over the ice sheet  indicate that melt never occurs.

[Figure]

**Figure S9.** The Antarctic ice sheet and  surrounding seas. Elevation contours are shown every 1000 m.

[Figure]

**Figure S10.** a: Mean specific humidity modelled by MAR over 1979–2015 at 700 hPa (Units: g/kg). Difference in mean specific humidity (%) between the reference simulation and (b) SST-4/SIC+6, (c) SST+4/SIC-6, (d) SST-2/SIC+3, (e) SST+2/SIC-3 experiments. Differences lower than the interannual variability are considered as non-significant and are dashed.

[Figure]

**Figure S11.** a: Mean air temperature modelled by MAR over 1979–2015 at 600 hPa (Units: °C). Difference in mean air temperature (°C) between the reference simulation and (b) SST-4/SIC+6, (c) SST+4/SIC-6, (d) SST-2/SIC+3, (e) SST+2/SIC-3 experiments. Differences lower than the interannual variability are considered as non-significant and are dashed.

[Figure]

**Figure S12.** Same as Fig. S11 but at 700 hPa.

**Table S1.** Top: Annual mean integrated (Gt yr$^{-1}$) and standard deviation (Gt yr$^{-1}$) total precipitation (rainfall and snowfall), snowfall and rainfall ver the whole AIS (including grounded and not grounded ice) for the reference simulation (1979–2015). Bottom: Difference of annual mean total precipitation (rainfall and snowfall), snowfall and rainfall (Gt yr$^{-1}$ and %) between each sensitivity test and the reference simulation (1979–2015). Anomalies larger than the inter-annual variability are considered as significant and are displayed in bold.

| Mean (Gt y$^{-1}$) | Total precipitation | Snowfall | Rainfall |
|---|---|---|---|
| Reference | 2678 ± 110 | 2658 ± 109 | 20± 3 |
| Anomaly (Gt y$^{-1}$) | Total precipitation | Snowfall | Rainfall |
| SST-4 | -64 | -61 | **-3** |
| SST-2 | -89 | -85 | **-4** |
| SST+2 | +50 | +45 | **+5** |
| SST+4 | **+162** | **+137** | **+25** |
| SIC+6 | **-170** | **-166** | **-4** |
| SIC+3 | -107 | -104 | **-3** |
| SIC-3 | +25 | +28 | **-3** |
| SIC-6 | +91 | +93 | -2 |
| SST-4/SIC+6 | **-136** | **-133** | **-3** |
| SST-2/SIC+3 | **-129** | **-125** | **-4** |
| SST+2/SIC-3 | **+133** | **+126** | **+7** |
| SST+4/SIC-6 | **+344** | **+304** | **+40** |
| SST/SIC-Nor | -105 | -102 | **-3** |
| SIC-CMIP | +36 | +35 | +1 |
| SST-CMIP | +80 | +79 | +1 |
| SST/SIC-CMIP | +105 | +104 | +1 |
| SST/SIC-GIS | **+368** | **+353** | **+15** |